# Mg²⁺-dependent conformational equilibria in CorA and an integrated view on transport regulation

**Nicolai Tidemand Johansen**[1†], **Marta Bonaccorsi**[2†], **Tone Bengtsen**[3,4†], **Andreas Haahr Larsen**[1,4], **Frederik Grønbæk Tidemand**[1], **Martin Cramer Pedersen**[1], **Pie Huda**[5], **Jens Berndtsson**[6], **Tamim Darwish**[7], **Nageshewar Rao Yepuri**[7], **Anne Martel**[8], **Thomas Günther Pomorski**[9,10], **Andrea Bertarello**[2], **Mark Sansom**[4], **Mikaela Rapp**[6], **Ramon Crehuet**[3,11], **Tobias Schubeis**[2*], **Kresten Lindorff-Larsen**[3*], **Guido Pintacuda**[2*], **Lise Arleth**[1*]

[1]Condensed Matter Physics, Niels Bohr Institute, University of Copenhagen, Copenhagen, Denmark; [2]Centre de RMN à Très hauts Champs de Lyon (UMR 5280, CNRS / Ecole Normale Supérieure de Lyon / Université Claude Bernard Lyon 1), University of Lyon, Villeurbanne, France; [3]Structural Biology and NMR Laboratory and Linderstrøm-Lang Centre for Protein Science, Department of Biology, University of Copenhagen, Copenhagen, Denmark; [4]Department of Biochemistry, University of Oxford, Oxford, United Kingdom; [5]Australian Institute for Bioengineering and Nanotechnology, The University of Queensland, Brisbane, Australia; [6]Department of Biochemistry and Biophysics, Center for Biomembrane Research, Stockholm University, Stockholm, Sweden; [7]National Deuteration Facility, Australian Nuclear Science and Technology Organization, Lucas Heights, Australia; [8]Institut Laue–Langevin, Grenoble, France; [9]Section for Transport Biology, Department of Plant and Environmental Sciences, University of Copenhagen, Frederiksberg, Denmark; [10]Department of Molecular Biochemistry, Faculty of Chemistry and Biochemistry, Ruhr University, Bochum, Germany; [11]CSIC-Institute for Advanced Chemistry of Catalonia (IQAC), Barcelona, Spain

**\*For correspondence:**
tobias.schubeis@ens-lyon.fr (TS);
lindorff@bio.ku.dk (KL-L);
guido.pintacuda@ens-lyon.fr (GP);
arleth@nbi.ku.dk (LA)

†These authors contributed equally to this work

**Competing interest:** The authors declare that no competing interests exist.

**Abstract** The CorA family of proteins regulates the homeostasis of divalent metal ions in many bacteria, archaea, and eukaryotic mitochondria, making it an important target in the investigation of the mechanisms of transport and its functional regulation. Although numerous structures of open and closed channels are now available for the CorA family, the mechanism of the transport regulation remains elusive. Here, we investigated the conformational distribution and associated dynamic behaviour of the pentameric Mg²⁺ channel CorA at room temperature using small-angle neutron scattering (SANS) in combination with molecular dynamics (MD) simulations and solid-state nuclear magnetic resonance spectroscopy (NMR). We find that neither the Mg²⁺-bound closed structure nor the Mg²⁺-free open forms are sufficient to explain the average conformation of CorA. Our data support the presence of conformational equilibria between multiple states, and we further find a variation in the behaviour of the backbone dynamics with and without Mg²⁺. We propose that CorA must be in a dynamic equilibrium between different non-conducting states, both symmetric and asymmetric, regardless of bound Mg²⁺ but that conducting states become more populated in Mg²⁺-free conditions. These properties are regulated by backbone dynamics and are key to understanding the functional regulation of CorA.

## Introduction

Magnesium is the most abundant divalent cation ($Mg^{2+}$) inside the cell, where it is mainly associated with the biological energy source adenosine triphosphate and other negatively charged molecules (*Jahnen-Dechent and Ketteler, 2012*). $Mg^{2+}$ serves several biological functions, for example as co-factor for enzymes (*Jahnen-Dechent and Ketteler, 2012*), and $Mg^{2+}$ deficiency is linked to severe diseases including cardiac syndromes, muscular dysfunction and bone wasting (*Rude, 1998*; *de Baaij et al., 2015*; *DiNicolantonio et al., 2018*). CorA is the main ion channel for $Mg^{2+}$-import in most bacteria and archea (*Maguire, 2006*). Despite little sequence conservation, CorA shares two membrane spanning helices and a conserved GMN motif with eukaryotic homologs, including Mrs2 that is responsible for $Mg^{2+}$-import to the mitochondrial lumen and is essential for cell survival (*Papp-Wallace and Maguire, 2007*; *Knoop et al., 2005*).

Several structures determined by X-ray crystallography are available for *Thermotoga maritima* CorA (TmCorA) (*Lunin et al., 2006*; *Eshaghi et al., 2006*; *Payandeh and Pai, 2006*; *Nordin et al., 2013*; *Pfoh et al., 2012*). All wild-type proteins have been crystalized as nearly symmetric pentamers in the presence of divalent metal ions and all represent a non-conducting state of the channel with a narrow and hydrophobic pore. *Figure 1A* shows a representative structure, which is characterized by a transmembrane domain (TMD) connected to the intracellular domain (ICD) by a long stalk helix. The periplasmic entrance to the pore contains the conserved GMN motif that presumably binds to $Mg^{2+}$ via its first hydration shell and thereby acts as a selectivity filter (*Nordin et al., 2013*; *Palombo et al., 2013*; *Dalmas et al., 2014a*). The ICD contains ten inter-protomer binding sites for $Mg^{2+}$ (two per protomer, denoted M1 and M2) involved in regulating the channel (*Nordin et al., 2013*; *Payandeh et al., 2008*; *Dalmas et al., 2014b*). The open state(s) of CorA have so far not been crystallized, but several biochemical and structural studies (*Payandeh et al., 2008*; *Dalmas et al., 2014b*) as well as molecular dynamics simulations (*Pfoh et al., 2012*) have pinpointed the determining residues involved in gating and suggested open models. One model suggests pore dilation upon loss of $Mg^{2+}$ at the M1 (and M2) sites due to a concerted iris-like movement (*Dalmas et al., 2014b*; *Chakrabarti et al., 2010*), while another suggests a hydrophobic-to-polar transition of the pore upon concerted rotation of the stalk helices (*Guskov et al., 2012*; *Kowatz and Maguire, 2019*).

Recently, cryo-EM structures were obtained both in the presence and absence of $Mg^{2+}$(*Matthies et al., 2016*). The $Mg^{2+}$-bound structure at ~3.8 Å resolution was symmetric and closed, in line with crystal structures, whereas two $Mg^{2+}$-free structures at ~7.1 Å were symmetry broken and with dilated pores. *Figure 1B* shows an intracellular view of the symmetric and asymmetric states, highlighting the symmetry break upon removing $Mg^{2+}$. From these observations, the proposed model involves a sequential destabilisation of CorA upon $Mg^{2+}$ removal, leading to a highly dynamic protein with shuffling protomers in the ICD, increasing the likelihood of pore dilation and wetting events (*Matthies et al., 2016*; *Neale et al., 2015*). Recent coarse-grained MD simulations revealed the residue level

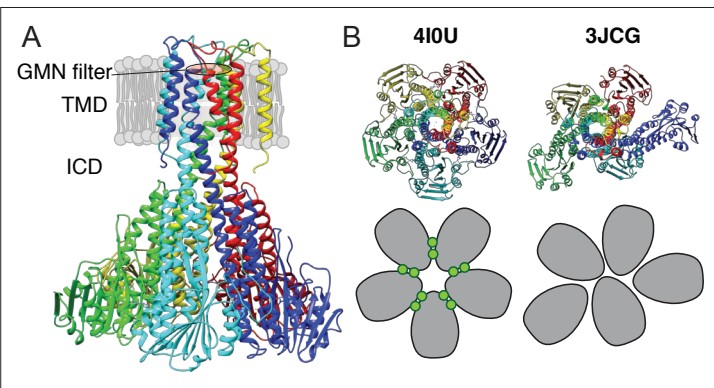

**Figure 1.** X-ray and cryo-EM structures of CorA. (**A**) Side view of symmetric CorA (PDB ID: 4I0U) in presence of $Mg^{2+}$ ('closed form'). (**B**) Top view of the same symmetric state of CorA (PDB ID: 4I0U) side-by-side to one of the asymmetric states observed in the absence of $Mg^{2+}$ ('open form') (PDB ID: 3JCG). A schematic representation of the two forms is shown below their structures, with each monomer shown in gray and $Mg^{2+}$ ions represented as green circles.

details of how a complex interaction network involving asymmetric movements of ICD monomers ultimately led to a conducting state upon removal of $Mg^{2+}$ (***Nemchinova et al., 2021***). High-speed atomic force microscopy (HS-AFM) data on densely packed CorA in lipid bilayers supported this model, but at the same time provided more insight to the dynamic interconversion of different states, including a fourth population of highly asymmetric CorA, not resolved by cryo-EM (***Rangl et al., 2019***). Interestingly, this population accounted for most observed conformations at low $Mg^{2+}$ concentrations, supporting that CorA is a dynamic protein with a relatively flat energy landscape and, potentially, multiple open states. However, CorA mutants with mutated regulatory M1 sites were still able to crystallize in the (symmetric) closed state (***Kowatz and Maguire, 2019***), suggesting that inter-protomer binding of $Mg^{2+}$ is not required for closing the channel. Overall, the cryo-EM and AFM experiments hint towards a highly dynamic ensemble of primarily asymmetric states at low $Mg^{2+}$ concentrations, while the successful crystallisation of M1 site mutants suggests that the closed state is significantly present at these conditions.

In this study, we investigated CorA using two room-temperature methods, namely small-angle neutron scattering (SANS), sensitive to large amplitude conformational changes and magic-angle spinning solid-state NMR (MAS NMR), sensitive to structure and dynamics with atomic resolution (***Reif et al., 2021***; ***Bonaccorsi et al., 2021***). For both methods, we employ custom-developed state-of-the-art methodology, that is size-exclusion chromatography (SEC) coupled to SANS (***Johansen et al., 2018***; ***Jordan et al., 2016***) and match-out deuterated carrier systems for SANS (***Maric et al., 2014***; ***Midtgaard et al., 2018***) (so-called stealth carrier systems), and >100 kHz MAS NMR in lipid bilayers (***Schubeis et al., 2018***; ***Schubeis et al., 2020***). Based on these data in conjunction with molecular simulations and modelling, we propose a model in which CorA is in a dynamic equilibrium between symmetric and asymmetric states, independent of bound $Mg^{2+}$, but where an ensemble of conducting states is energetically more favourable for $Mg^{2+}$-free CorA due to increased conformational dynamics resulting from the released electrostatic constraint.

## Results

### CorA is structurally similar in presence and absence of $Mg^{2+}$

The published cryo-EM structures of CorA in absence of $Mg^{2+}$ (***Figure 1B***, 3JCG) reveal large structural rearrangements compared to the nearly symmetric, non-conductive state obtained from crystallography (***Figure 1B***, 4I0U). SANS curves calculated from these two structural states of CorA reveal a significant change in the scattering curve in the region $q$ = 0.08 Å$^{-1}$ – 0.15 Å$^{-1}$ (***Figure 2A and B***, right panels), that is on a length scale that is well-covered in a standard SANS experiment. To match the cryo-EM conditions, we performed SANS measurements in n-dodecyl-B-D-maltoside (DDM) detergent micelles and 2-Oleoyl-1-palmitoyl-sn-glycero-3-phosphocholine (POPC) lipid nanodiscs. We used selectively deuterated versions of both carrier types that were homogenously matched-out and hence invisible at 100% $D_2O$; that is stealth DDM (sDDM) and stealth nanodiscs (sND, ***Figure 2— figure supplement 1***). Strikingly, the measured SANS curves are pair-wise indistinguishable in the absence of $Mg^{2+}$ (1 mM EDTA) and in the presence of 40 mM $Mg^{2+}$ for the sDDM (***Figure 2A***) and sND (***Figure 2B***) samples, respectively, indicating no significant difference in the average conformations of the $Mg^{2+}$-free and bound states of CorA. This observation contrasts with the recently proposed large-scale structural rearrangements reported from cryo-EM and high-speed AFM data.

We note that the SANS data obtained on the sND samples (***Figure 2B***) have a slight excess scattering contribution at low-$q$ compared to the sDDM samples (***Figure 2A***), which can be attributed to the presence of a few *E. coli* endogenous lipids in the sND samples. However, the SANS data from sDDM and sND samples are indistinguishable in the $q$-region expected to reveal differences from symmetric and asymmetric states (***Figure 2—figure supplement 2***), which confirms that CorA exhibits the same behaviour in a POPC lipid environment and in DDM detergent carriers. The rightmost panels of ***Figure 2A and B*** show enhanced views on this region for the SANS data compared to the SANS curves calculated from the PDB structures. Interestingly, neither of the curves calculated from the PBD structures match the measured SANS data, suggesting that the solution structure of CorA cannot be described by any of these single structures, regardless of whether or not CorA is in the presence of $Mg^{2+}$.

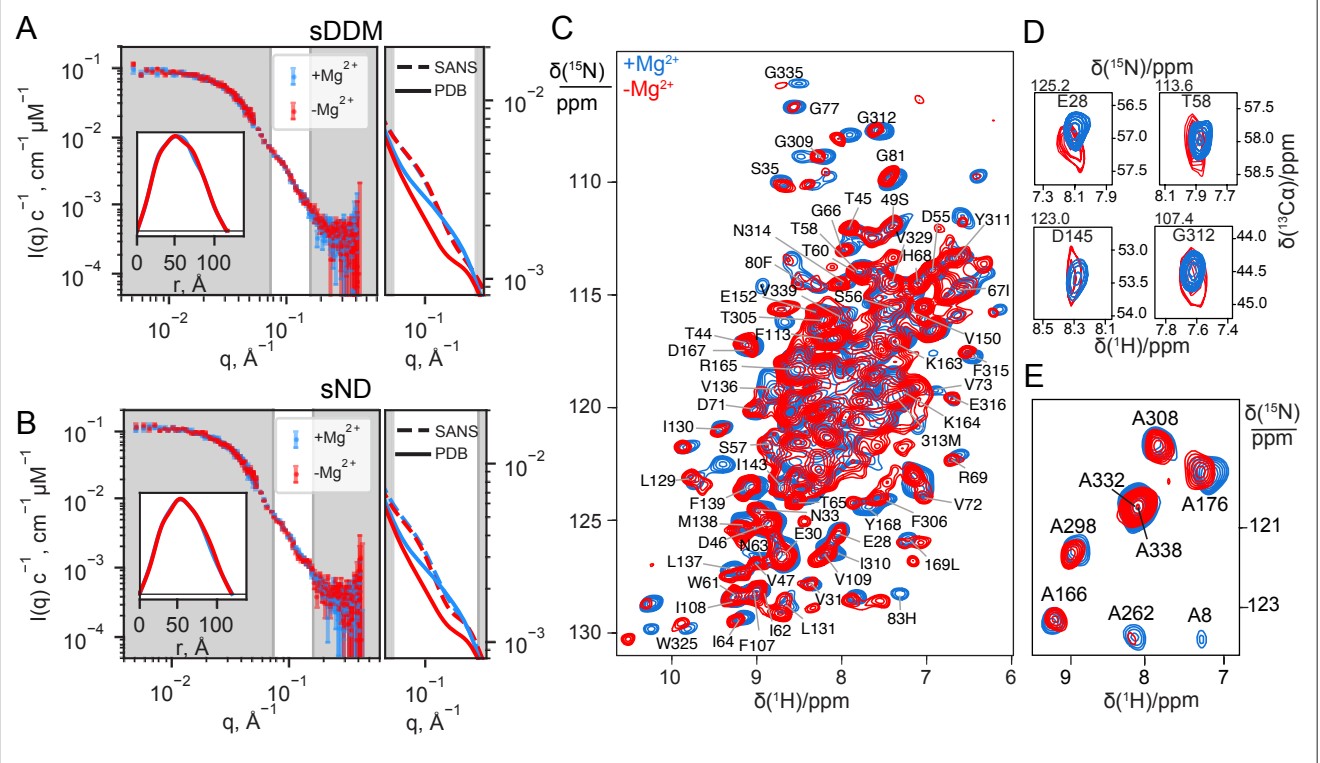

**Figure 2.** Experimental data on CorA in presence (blue) and absence (red) of Mg$^{2+}$. (**A + B**) Experimental SANS data of CorA embedded in stealth DDM micelles (sDDM) and stealth nanodiscs (sND), respectively, with p(r)-distributions calculated on BayesApp (*Hansen, 2014*) in the inset. The rightmost plots show zoomed comparisons of the p(r)-fits of experimental data (SANS, dashed lines) with the SANS curves calculated on the X-ray (4I0U) and cryo-EM (3JCG) structures (PDB, full lines). Complete fits based on the PDB structures are shown in *Figure 4A*. (**C**) 2D $^1$H-$^{15}$N dipolar correlation spectra by MAS NMR of CorA in hydrated DMPC bilayers recorded at 1 GHz $^1$H Larmor frequency and 107 kHz MAS. (**D**) 2D sections for a selection of residues obtained from 3D $^1$H-$^{15}$N-$^{13}$Cα spectra recorded at 1 GHz $^1$H Larmor frequency and 107 kHz MAS. (**E**) 2D $^1$H-$^{15}$N dipolar correlation spectra obtained for $^{15}$N-Alanine-labeled CorA recorded at 1 GHz $^1$H Larmor frequency and 60 kHz MAS. In C and E, site-specific assignments are annotated for resolved resonances.

The online version of this article includes the following figure supplement(s) for figure 2:

**Figure supplement 1.** Validation of nanodisc match-out deuteration.

**Figure supplement 2.** Comparison of SANS data in sDDM and sND.

**Figure supplement 3.** Absolute chemical shift differences in 3D $^1$H-$^{15}$N-$^{13}$Cα spectra of CorA with and without Mg$^{2+}$, calculated as Δ(δ)=
$$\sqrt{\left(\delta_{X,+Mg} - \delta_{X,-Mg}\right)^2}$$ for X=$^1$H,$^{13}$Cα,$^{15}$N.

While SANS data provided information on the overall molecular shape of CorA in the two preparations, we used MAS NMR to obtain insight into structural changes at the residue-level length-scale. MAS NMR data were recorded on uniformly $^{13}$C,$^{15}$N-labelled CorA, reconstituted in 1,2-dimyristoyl-sn-glycero-3-phosphocholine (DMPC) lipid bilayers, in the presence or absence of Mg$^{2+}$. Backbone resonance assignment was obtained at high Mg$^{2+}$ concentration acquiring a set of three-dimensional experiments relying on $^1$H$^N$ and $^1$Hα detection with 100 kHz MAS. We were able to annotate ~100 peaks to residues spread throughout the structure of CorA. Notably, the assignment of CorA with and without Mg$^{2+}$ is clustered in the globular region in the ICD and in the TMD, including the important periplasmic loop, whereas only sparse assignments were established in the long portion (243-289) of the stalk helix connecting the two regions. The determination of random coil chemical shift deviation (CSD) values confirmed that the secondary structure is in good agreement with the one obtained by X-ray crystallography and Cryo-EM.

2D $^1$H-$^{15}$N dipolar correlation spectra represent direct structural 'fingerprints' of CorA in the two preparations. Despite the high signal overlap associated to the high-molecular-weight of CorA, fast MAS rates and the ultra-high magnetic field guarantee high sensitivity and feature numerous signals with a resolution sufficient to track subtle structural changes. Once again, against our expectations,

we remarked that the spectra with and without $Mg^{2+}$ showed very little difference, with the positions of the resolved peaks differing less than 0.1 ppm in the two forms and without peak splitting or broadening that would indicate distinct conformations (*Figure 2C*).

Two parallel strategies were pursued to extend the analysis to the more crowded regions. First, we acquired three-dimensional (3D) experiments which correlate the amide proton and nitrogen with the $C_\alpha$-carbon within each residue and thus include an additional $^{13}C$ chemical shift dimension (*Figure 2D*). The resulting 3D spectra confirmed negligible chemical shift variations over more than ~90 sites across the TMD and ICD (*Figure 2D*, *Figure 2—figure supplement 3*). In the $Mg^{2+}$-free sample, however, a notable decrease in signal intensity was observed for most residues, resulting in the complete disappearance of two thirds of the peaks from the TMD (vide infra).

Secondly, we used amino-acid-specific isotopic enrichment to select the NMR signals associated to the amide groups of the alanine residues. Each CorA protomer contains eight alanine residues, distributed with four in the TMD and four in the ICD, which were all visible and assigned in the corresponding $^1H$-$^{15}N$ dipolar correlation spectra of two preparations with and without magnesium (*Figure 2E*). Also, in this case, the spectra are superimposable and show no evidence of peak splitting.

In conclusion, and in line with SANS, the NMR data show that the predominant structure of CorA in lipid bilayers is unaltered by the removal of $Mg^{2+}$.

## CorA is active and preserves its tertiary structure in $D_2O$

Since $Mg^{2+}$ hydration plays an important role in CorA selectivity, and $D_2O$ and $H_2O$ have slightly different physicochemical properties (*Némethy and Scheraga, 1964*), we speculated whether the identical SANS curves with and without $Mg^{2+}$ were due to CorA losing its activity in the SANS condition, that is at 100% $D_2O$. To test this, we measured the activity of CorA in POPC liposomes under the SANS conditions by a fluorometric assay (*Figure 3A*). This analysis shows that CorA is clearly active in $D_2O$. The transport rate estimated from the initial linear part of the trace is lower by less than a factor of two compared to $H_2O$. A slightly reduced rate in $D_2O$ has been reported for other membrane proteins (*Sugiyama and Yoshiok, 2012*) and is explainable by slightly altered properties of the two solvents (*Némethy and Scheraga, 1964*; *Hummer et al., 2000*). We could also inhibit CorA activity in $D_2O$ with $Co[NH3]_6^{3+}$ (*Figure 3A*, green), supporting that the protein is indeed functional under the SANS conditions.

We further carried out negative stain EM on samples of CorA in DDM and $D_2O$ with or without $Mg^{2+}$. The refined 2D classes clearly show that the pentameric architecture of CorA is preserved in $D_2O$ in both conditions (*Figure 3B*). Furthermore, several 2D classes appear to exhibit approximate fivefold

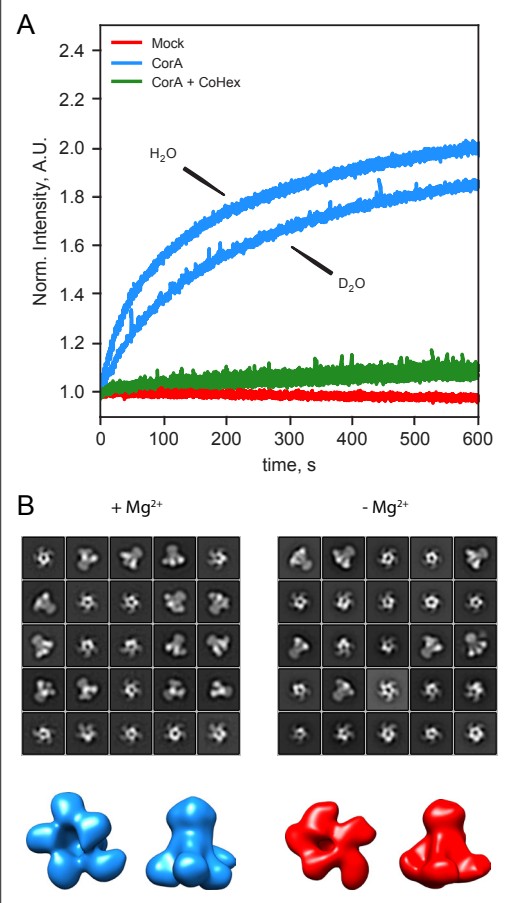

**Figure 3.** CorA activity in $D_2O$ and direct visualization by negative stain EM. (**A**) CorA activity in the conditions used for SANS. The traces are the normalized fluorescence signals after adding $Mg^{2+}$ to either empty POPC liposomes (Mock), CorA-POPC proteoliposomes (CorA), or CorA-POPC proteoliposomes preincubated with the inhibitor $Co[NH_3]^{3+}$ in $D_2O$ (CorA+ CoHex). (**B**) Negative stain EM of CorA in DDM and $D_2O$ with and without $Mg^{2+}$. The 25 most abundant 2D classes are shown for each condition. The box dimensions are 170 × 170 $Å^2$ for scale. (**C**) Final 3D model for each condition shown from the intracellular side and in side-view, respectively.

symmetry, both in the presence and absence of $Mg^{2+}$. To avoid bias, we refined 3D models without imposing symmetry (*Figure 3C*). In both conditions, these low-resolution 3D models ($\approx$ 15 Å) are reminiscent of the overall expected architecture of CorA (*Figure 1*), but notably show some degree of asymmetry. In conclusion, CorA show little to no perturbation from measurements in $D_2O$, rendering the SANS data sets viable for modelling of the solution structure of CorA.

## Model refinement to SANS data shows that CorA is asymmetric

The SANS data sets obtained in sDDM (*Figure 2A*) exhibit well-defined Guinier-regions and the calculated radii of gyration, $R_g$, of 42.1 ± 1.3 Å (+ $Mg^{2+}$) and 43.8 ± 1.7 Å (- $Mg^{2+}$) are close to the predicted values from the X-ray (41.3 Å) and cryo-EM structures (42.2 Å and 42.3 Å). Thus, these data are indicative of well-separated CorA pentamers with no interference from visible lipids or the kind, providing the optimal basis for structural modelling. Given the high similarity of the SANS data obtained on the samples of CorA in sDDM with and without $Mg^{2+}$, we performed structural modelling on only a single data set, that is CorA in sDDM without $Mg^{2+}$ (*Figure 2A*, red). Despite controversies on the open state, there is consensus that the crystallized symmetric state represents the closed state of the protein. Surprisingly, we could not obtain good fits of the symmetric state to our SANS data without clear systematic deviations, especially at the feature present at $q \approx 0.1$ Å$^{-1}$ (*Figure 4A*, 4I0U). This was also the case for the asymmetric cryo-EM structure (*Figure 4A*, 3JCG) that produced an even worse fit. In SANS, the signal represents a population-weighted average of all conformations that the protein can adopt. With a measurement time on the order of several minutes and illumination of $\approx 10^{12}$–$10^{13}$ molecules, all accessible populations are expected to contribute to the signal. A relatively flat energy landscape with multiple interconverting states has been proposed in $Mg^{2+}$-free conditions, making a fit of a single structure less meaningful in this context. However, it is unlikely that the average of an ensemble of asymmetric structures give rise to the same SANS signal as that of a single symmetric state corresponding to the structure determined by crystallography and cryo-EM in high $Mg^{2+}$.

With no scattering contribution from the carrier systems, it becomes possible to analyse the SANS data by conventional methods for soluble proteins, such as bead-modelling. When imposing P5 symmetry, we could obtain envelopes reminiscent of the CorA structure by bead-modelling (*Figure 4—figure supplement 1B*), whereas no symmetry (P1) imposed led to asymmetric mass distributions that were not at all compatible with the overall architecture of CorA (*Figure 4—figure supplement 1A*). This indicates that there is significant structural dispersion in the sample.

To obtain a molecular constrained model compatible with the data, we applied a modified type of normal mode analysis (NMA), starting from the closed crystal structure. A structure with mostly intact secondary structure but a high degree of asymmetry in the ICDs yielded a good fit to the SANS data (*Figure 4B*). Importantly, this model describes the feature at $q \approx 0.1$ Å$^{-1}$, where the PDB models deviate the most. Thus, on average, the solution structure of CorA appears to be asymmetric, in line with our EM models (*Figure 3C*). Again, we emphasize that such an overall asymmetric structure of CorA in presence of excess $Mg^{2+}$ is in stark contrast to the picture of a closed, symmetric structure that has served the basis for all proposed mechanisms of $Mg^{2+}$ gating. However, a single asymmetric model as derived from NMA (*Figure 4B*) is neither compatible with a single set of peaks in NMR (*Figure 2C and E*) nor the substantial experimental evidence for a nearly symmetric, closed state in presence of $Mg^{2+}$. Likely, CorA adopts multiple different states (*Matthies et al., 2016*; *Rangl et al., 2019*), and according to our data does so both with and without $Mg^{2+}$ bound. In this case, the SANS data would represent the number-weighted average of different states that must be overall asymmetric.

To model the apparent asymmetry in CorA in more detail and as a conformational ensemble, we performed coarse grained molecular dynamics simulations (MD). First, we set up CorA embedded in a POPC bilayer using the Martini3.0b force field. Starting from the symmetric or asymmetric structures, 32 µs and 20 µs simulations, respectively, without any inter-chain elastic network terms yielded only small structural fluctuations, which did not significantly improve fits to the experimental SANS data, especially not around the feature at $q \approx 0.1$ Å$^{-1}$. Thus, we extended the analysis to metadynamics simulations (MetaD) that allows for enhanced sampling of structural dynamics. MetaD drives the simulation towards a larger variety of structural states based on an energetic bias on a structural feature, a so-called collective variable, here the $R_g$ on specific ICD residues. Starting from the symmetric structure, the MetaD simulation quickly drove the simulation away from the local structural minimum that

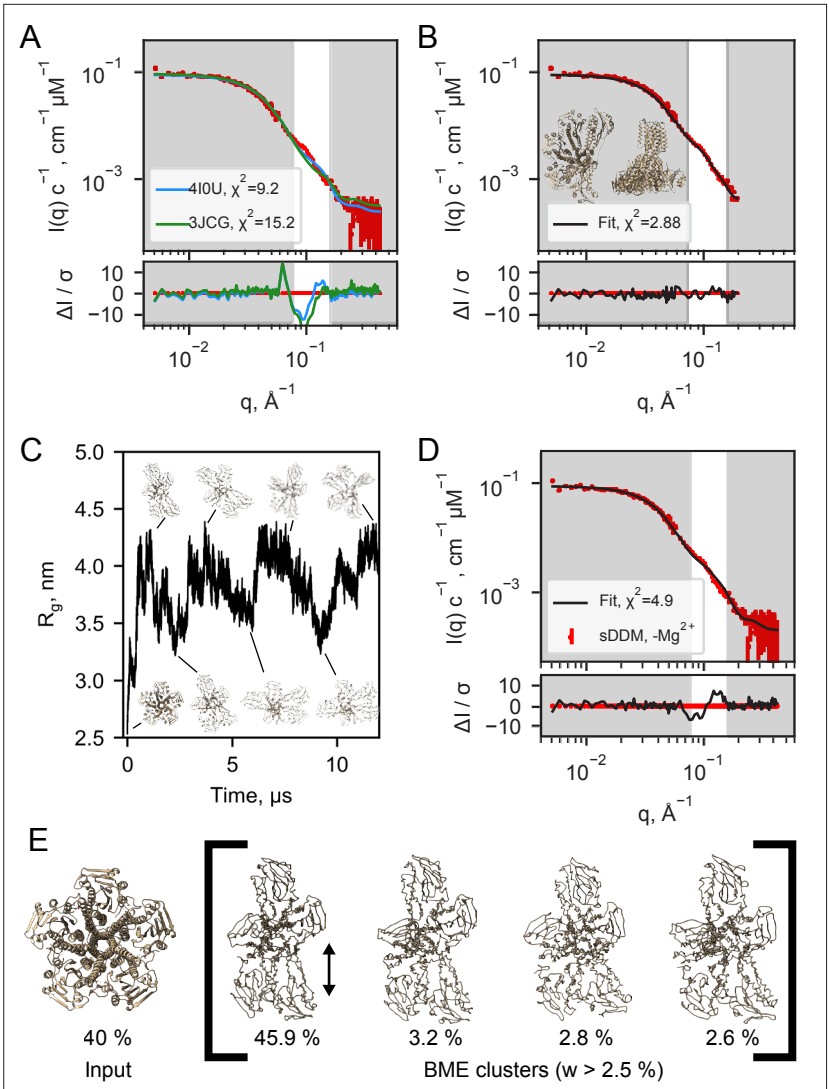

**Figure 4.** Structural modelling of CorA from SANS data. (**A**) Model fits of the closed crystal structure (4I0U) and open cryo-EM structure (3JCG) to the experimental SANS data obtained in sDDM without $Mg^{2+}$. The bottom panel shows the error-normalised difference plot of the fits. (**B**) Model fit of the structure obtained by regularised normal mode analysis (bottom and side views inserted on plot). (**C**) MetaD molecular dynamics trajectory with representative frames visualized as ribbon structures. (**D**) Fit of a weighted ensemble obtained from the MD simulation in B together with a 40% contribution from the symmetric CorA. (**E**) The four highest weighted cluster centroids in the metaD simulation (right bracket) together with the closed symmetric structure (left) which together illustrates the ensemble of CorA structures that produced the best fit to the experimental SANS data.

The online version of this article includes the following figure supplement(s) for figure 4:

**Figure supplement 1.** Bead modelling of CorA.

**Figure supplement 2.** Goodness of fit from NMA generated structures to SANS.

**Figure supplement 3.** Ensemble fitting to SANS data with symmetric CorA included.

**Figure supplement 4.** Representation of the pore radius profile in the transmembrane domain obtained by HOLE program out of the structures calculated by Normal Mode Analysis and MetaDynamics.

the standard MD had been trapped in and sampled a large range of conformationally different structures (*Figure 4C*).

The averaged back-calculated SANS of the entire ensemble of structures obtained from MetaD simulations did not fit the experimental SANS data satisfactorily. This can be explained by for example remaining inaccuracies in the coarse grained force field and imprecisions in the simulation

from insufficient sampling. To resolve this, we applied the Bayesian/Maximum Entropy reweighting method (BME) to optimize the weights of the individual conformations in the simulation with the aim of obtaining an ensemble in better agreement with the experimental SANS data. In addition to the BME, we enforced that a symmetric state should be present in the final ensemble, given the substantial experimental evidence for this state in the literature and that it was under-sampled in the simulation. The best fit to the SANS data (*Figure 4D*) was obtained with an ensemble consisting of 40% ± 28% symmetric CorA structure and the remainder of asymmetric conformations from the MetaD (*Figure 4E* and *Figure 4—figure supplement 3*). Despite some systematic deviations, the fit is much improved with regards to describing the feature in the SANS data at $q \approx 0.1$ Å⁻¹, as compared to the fits obtained with the symmetric crystal structure or the open cryo-EM structures, respectively (*Figure 4A*).

To visualise the results of the MetaD simulation and hence the simulated dynamics, we cluster similar structures and show the four most predominant cluster centroids (*Figure 4E*) where especially one cluster contributes to the final fitted ensemble. Although the four cluster centroids are wide apart in simulation time (≈500 ns), they are structurally similar with a maximum pairwise RMSD of 6.5 Å (data not shown). The main difference is the distance between the two protomers (*Figure 4E*, black arrow), which indicates that the individual domains of the ICD can move relative to each other. This is in line with the subunit displacements described in the symmetry-break-based gating model derived from the cryo-EM structures and supported by AFM measurements. Importantly, however, we find that these movements occur irrespective of the presence of $Mg^{2+}$, given the nearly identical SANS data in the two conditions.

To investigate whether asymmetric transitions correspond to an opening of the channel, we analysed critical distances between protomers within the NMA- and metaD-generated structures. In the asymmetric structures refined from MetaD simulations, only a single binding site in the cytoplasmic region remains competent to coordinate $Mg^{2+}$, whereas in the NMA-generated structure, four binding sites remain within distance to bind $Mg^{2+}$. In the transmembrane region, the pore dimensions obtained by the HOLE program (*Smart et al., 1996*) indicate that for all the NMA- and MetaD-generated structures, the pores contain narrow constrictions with diameters significantly smaller than the 2.4 Å radius of hydrated $Mg^{2+}$ (*Maguire and Cowan, 2002*, *Figure 4—figure supplement 4*). This suggests that all the refined asymmetric structures are in fact non-conducting. Taken together, the general picture of our simulations does not provide a mechanistic link between symmetry break and channel-opening. On the other hand, this suggests that symmetry-break alone is not sufficient for obtaining ion transport.

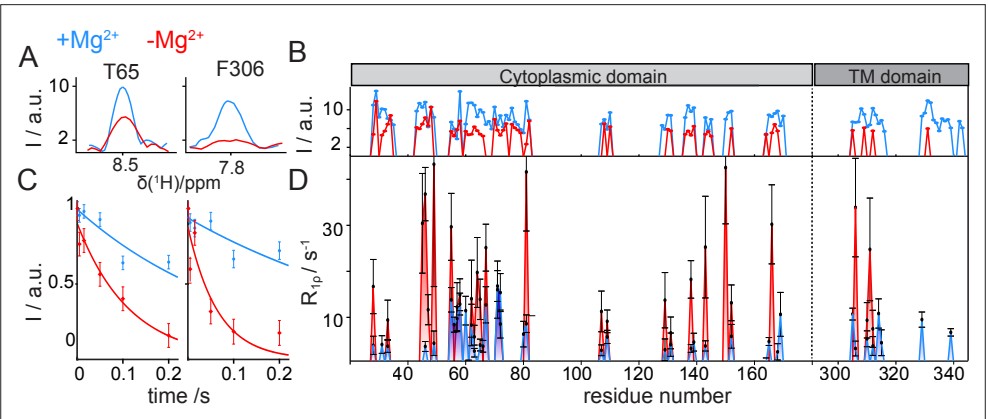

**Figure 5.** CorA backbone dynamics by MAS NMR in DMPC in presence (blue) and in absence (red) of $Mg^{2+}$. (**A**) Examples of 1D traces of 3D ¹H-¹⁵N-¹³Cα peaks for two residues in the ICD (T65) and in the TMD (F306). (**B**) Comparison of peak intensities in 3D ¹H-¹⁵N-¹³Cα spectra over the protein sequence. ICD and TMD are indicated by boxes of different color on top of the plot. C-D: Site-specific ¹⁵N $R_{1\rho}$ rates measured with a 15 kHz spin-lock field. (**C**) Examples of ¹⁵N $R_{1\rho}$ relaxation decays together with the corresponding mono-exponential fits for residues T65 and F306. (**D**) Comparison of site-specific backbone ¹⁵N $R_{1\rho}$ rates plotted along the CorA sequence.

The online version of this article includes the following figure supplement(s) for figure 5:

**Figure supplement 1.** CorA backbone dynamics by MAS NMR: Site-specificity and temperature dependence.

## Structural dynamics are different in open and closed states of CorA

So far, we have considered a set of static snapshots to interpret the wide variety of populated states of CorA. The observation of MAS NMR dynamical probes sheds light on the backbone motions of these states over different timescales, enriching the structural description of CorA with conformational plasticity. A first insight on site-specific dynamics is obtained by comparing the peak intensities observed in the MAS NMR experiments in the two samples with and without $Mg^{2+}$. Peak intensities are dependent on dipolar couplings between nearby nuclei and are affected when such couplings are averaged by local motions. For an amide $^1H$-$^{15}N$ pair, this corresponds to motional processes faster than tens of kHz (i.e. more rapid than hundreds of µs).

Changes in peak intensity are already noticeable in the 2D $^1H$-$^{15}N$ dipolar correlation spectra, but are amplified in the 3D correlations, where additional radio-frequency irradiation periods act as a more stringent filter, dumping the signals of the most mobile sites. 1D traces of two exemplar 3D $^1H$-$^{15}N$-$^{13}C\alpha$ correlations (T65 in the ICD and F306 in the TMD) and the plot of signal intensities over the full protein sequence with and without $Mg^{2+}$ are shown in *Figure 5A and B*, respectively. As mentioned above, an overall decrease in peak intensities is associated to removal of $Mg^{2+}$, with a stronger effect observed in the TMD. This points toward a variation of the dynamic behavior of this region in the two samples.

$^{15}N$ spin-lattice relaxation rates in the presence of a spin-lock field ($^{15}N$ $R_{1\rho}$) are sensitive reporters of motions occurring over a window of hundreds of ns to hundreds of µs. These parameters were measured site-specifically for ~40 residues in the CorA backbone, by introducing a relaxation filter in the $^1H$-$^{15}N$ dipolar correlation module and monitoring the signal decay of each amide pairs in a series of experiments with increasing relaxation delays (*Figure 5C*). Despite the intrinsically low signal-to-noise ratio of the experiments, which results in very large uncertainties on the site-specific relaxation rates, a remarkable (up to 10 fold) systematic increase was observed throughout the entire protein upon removal of $Mg^{2+}$ (*Figure 5D*). The same effect was observed for the $^{15}N$-alanine labelled sample (*Figure 5—figure supplement 1A*). An additional measurement of $^{15}N$ $R_{1\rho}$ rates upon a ~ 20 °C cooling revealed a different temperature-dependent behaviour of the two $Mg^{2+}$-loaded samples (*Figure 5—figure supplement 1B*). While in the presence of $Mg^{2+}$ relaxation rates were conserved at low temperatures, an important (2–3 fold) decrease was observed in the $Mg^{2+}$-free form. Finally, also dynamics on the ns timescale obtained by the measurements of bulk backbone $^{15}N$ $R_1$ showed a global increase of relaxation rates (by a factor ~1.5) upon removal of $Mg^{2+}$ (*Figure 5—figure supplement 1C*).

In summary, MAS NMR reveals that removal of $Mg^{2+}$ triggers an increase in the backbone flexibility on different timescales, a dynamical effect which is different for the ICD and the TMD.

## Discussion

The functional mechanism of the pentameric divalent cation channel CorA of *Thermotoga maritima* has been under investigation since the release of the first high-resolution structures in 2006. The gaiting mechanism is most commonly explained by a simple two-state model, involving a conformational transition between a closed symmetric state and one or many open asymmetric states. Indeed, the concise current model derived from cryo-EM (*Matthies et al., 2016*) and HS-AFM (*Rangl et al., 2019*) defines a rigid symmetric conformation at high $Mg^{2+}$ concentrations ( > 20 mM), dynamic asymmetric conformations at low $Mg^{2+}$ concentrations (2–3 mM) and several distinct rigid asymmetric conformations in the absence of $Mg^{2+}$.

We have performed SANS in stealth carrier systems to study the solution conformation of CorA, together with modeling of both static and ensemble structures obtained from MetaD simulations. Furthermore, we used $^1H$-detected solid-state NMR to investigate CorA conformation and dynamics in hydrated lipid bilayers. These complementary methods allow us to expand the current view on the mechanistically important conformational equilibria.

## Symmetric and asymmetric structures are populated both in the presence and absence of $Mg^{2+}$

While our SANS data probes the overall solution conformation of CorA at low resolution of around 20 Å, it is important to note that the technique is indeed quite sensitive to structural rearrangements

on much smaller length scales when the same sample, but under different conditions, is compared internally. As such, given the identical SANS data recorded in absence and presence of high $Mg^{2+}$ concentrations, we conclude that the overall solution structure of CorA is similar in both conditions. Our refinement of models to the SANS data shows that CorA is overall asymmetric, contradicting the idea of a single rigid, symmetric structure in high $Mg^{2+}$ concentrations. Indeed, our data can only be explained by symmetry breaking and/or a mixture of populations, for example through the concomitant presence of symmetric and asymmetric states. Furthermore, NMR spectra are sensitive to the environments of each nucleus, and thus report on conformational transitions breaking the local symmetries. The fact that NMR chemical shifts do not change significantly upon removal of $Mg^{2+}$ indicates the conservation of local symmetric environments for most of the NMR-active nuclei in both conditions. In turn, this evidence suggests that the global symmetry is maintained for a substantial population of CorA pentamers in the absence of $Mg^{2+}$. Hence, we have from SANS that symmetry breaking is present at both high and low $Mg^{2+}$, and from NMR that a substantial population of CorA is on the symmetric form in both $Mg^{2+}$ conditions. These findings are supported by our negative-stain TEM data, but also by the literature. For example, the symmetric cryo-EM structure (3JCF) obtained at high $Mg^{2+}$ concentrations, resulted from roughly 60% of the particles (*Matthies et al., 2016*), which implies that asymmetric states could partly account for the remaining 40%. The two asymmetric cryo-EM structures were refined from only about 15% each of the picked particles, indicating the presence of various conformations, and likely including the symmetric form. AFM provided a more detailed conformational analysis and found around 20% symmetric structures at low $Mg^{2+}$ concentrations (0–3 mM) (*Rangl et al., 2019*).

## Increased dynamics of CorA in absence of $Mg^{2+}$

An elevated conformational plasticity at low levels of $Mg^{2+}$ was first postulated due to the unsuccessful crystallization trials in these conditions. Structural evidence for this arrived with the finding of several asymmetric states on cryo-EM grids and was recently supported by the observation of real time dynamics on a timescale of seconds through high-speed AFM (*Rangl et al., 2019*). MAS NMR has the exclusive advantages of probing molecular motions with site-specific resolution and the possibility to tune the experiments towards a dynamic timescale of interest. On a fast ps-ns timescale, we observed increased bulk $^{15}N$ $R_1$s in the absence of $Mg^{2+}$, indicating less restricted backbone motions, as also previously suggested by MD simulations (*Chakrabarti et al., 2010*). Site-specific $^{15}N$ $R_{1\rho}$ rates

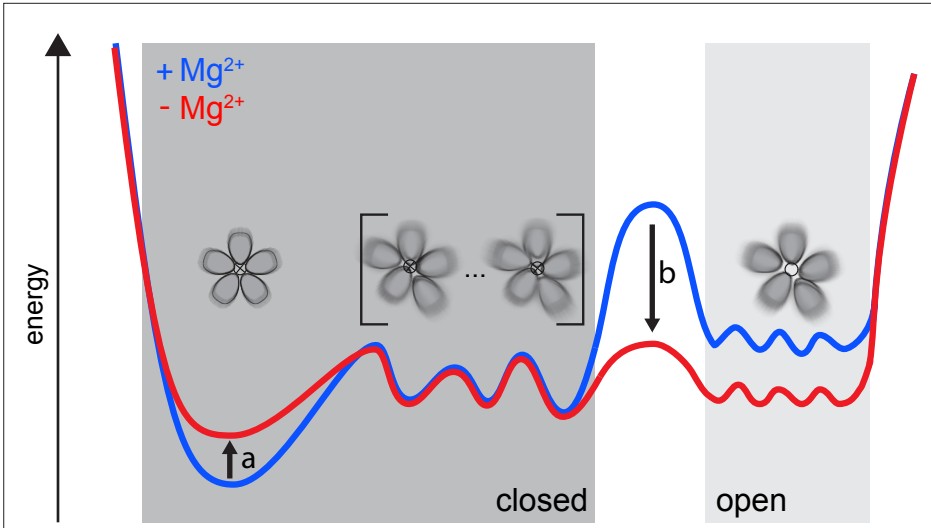

**Figure 6.** Our proposed dynamic model for CorA. Both in high and low intracellular $Mg^{2+}$ levels, a complex ensemble of closed states (left), symmetric as well as asymmetric, are in equilibrium with an ensemble of open states (right). At high $Mg^{2+}$(blue) the large energy barrier prevents CorA from visiting the ensemble of open states. At low intracellular $Mg^{2+}$ levels (red) a reduction in the population of the symmetric, closed state (arrow a) is induced. At the same time the energy barrier (arrow b) is lowered toward the ensemble of open states, which then becomes populated.

are reporters of segmental motions, for example displacement of secondary structure elements with respect to each other, over a window of hundreds of ns to hundreds of µs. These dynamical processes appear to be largely promoted over the whole CorA structure when $Mg^{2+}$ is absent. We also observed that motions in stalk helix 7, already present in the $Mg^{2+}$-bound sample, propagated into the transmembrane domain when $Mg^{2+}$ was removed. Thus, we can indeed confirm that the dynamics of CorA increase at low $Mg^{2+}$ concentrations, but also that motions occur on various timescales. A previous work suggested that removal of $Mg^{2+}$ resulted in a combination of lateral and radial tilting of two adjacent monomers, which allowed the creation of interactions between them (*Pfoh et al., 2012*). This idea was recently extended based on coarse-grained MD simulations that showed an ensemble of conformational changes propagating from the ICD to the TMD helices (*Nemchinova et al., 2021*). In line with these observations, we propose that collective ns-µs motions of the backbone initially promoted in the ICD by release of $Mg^{2+}$ ions would in turn induce higher conformational flexibility in the TMD. While a detailed understanding of those complex dynamic processes will require elaborate follow-up experiments, we can interpret our current findings in the view of how conformational changes regulate the channel opening.

## An integrated view on CorA transport regulation

First, we remark that our findings cannot be explained in terms of sharp transitions between open and closed states. In other words, a high degree of asymmetry is not equivalent to an open state. We propose an alternative model, schematically illustrated in *Figure 6*, that integrates our findings with the previously available literature. CorA samples symmetric and asymmetric conformational states, whose distribution is tuned by the $Mg^{2+}$ concentration. Between the symmetric, closed state and an ensemble of open states, a relatively flat energy landscape exists with asymmetric, closed states, both in the presence and absence of $Mg^{2+}$. Low $Mg^{2+}$ intracellular levels induce a reduction in the population of the symmetric, closed state together with a decrease of the energy barrier toward an ensemble of open states, which become populated. In this context, increased dynamics, as observed by NMR, can become the key determinant allowing CorA to explore different wells of the energy profile, making the open state reachable.

Such a dynamic model is compatible with previously unexplainable symmetric crystal structures of M1-binding site mutants in the absence of $Mg^{2+}$ (*Kowatz and Maguire, 2019*). While for WT CorA, removal of $Mg^{2+}$ increases the dynamics and shifts the conformational equilibria, point mutations probably have an opposite effect, stabilizing the symmetric state and allowing crystal formation. Furthermore, this model aligns with a previous MD study, which found that dry and intermediate transiently wetted states, both non-conducting, were present irrespective of bound $Mg^{2+}$ and with interconversion rates on the ns time scale, whereas without $Mg^{2+}$ bound, a conducting 'stably-superhydrated' state was sampled as well (*Neale et al., 2015*).

Intriguingly, the interplay between symmetric and asymmetric conformations have also been investigated for another family of pentameric channels, the ligand-gated ion channels (pLGIC) (*Rao et al., 2021*). While these are usually symmetric or pseudo-symmetric in the ligand-free form, studies suggest that asymmetric resting states exist as intermediates in the conformational landscape, and that they become more energetically favorable when substrate is bound (*Zhang et al., 2021*). As such, it appears that asymmetry might have a more widespread occurrence in channel regulation.

## Conclusion

The availability of room temperature SANS and NMR data of CorA in lipid bilayers, reporting on both global and local behaviour of this channel, supported by MD simulations, allowed us to extend and rationalise the 'symmetry-break-upon-gating' model for $Mg^{2+}$ transport. Our observations support the suggestion that asymmetric conformations are involved in the gating mechanism, but in a more complex way than a simpler two-state picture, where $Mg^{2+}$-bound CorA is a stable, symmetric structure. Indeed, we find that the CorA pentamer is a symmetry-broken fluctuating structure able to explore a wide conformational landscape both in the presence and absence of $Mg^{2+}$. We propose that the determining factor for CorA to visit conducting states is the increase in backbone flexibility on different timescales upon release of regulatory $Mg^{2+}$. Future investigations of the conformational equilibria will enrich the insight provided in this study with a better mechanistic understanding of the

kinetics and thermodynamics of gating and transport, and more detailed atomic-resolution simulations may shed light on the ion regulation and transport pathways.

# Materials and methods

## Key resources table

| Reagent type (species) or resource | Designation | Source or reference | Identifiers | Additional information |
|---|---|---|---|---|
| Gene (*Thermotoga maritima*) | corA | Uniprot | Q9WZ31 | |
| Strain, strain background (*Escherichia coli*) | BL21 star(DE3) | ThermoFisher | C601003 | |
| Software, algorithm | PEPSI-SANS | https://team.inria.fr/nano-d/software/pepsi-sans/ | RRID:SCR_021950 | |
| Software, algorithm | NOLB | Ref. 48 | RRID:SCR_021954 | |
| Software, algorithm | Martini3.0b | Ref. 49 | RRID:SCR_021951 | |
| Software, algorithm | GROMACS-5.1.4 | Ref. 52 | RRID:SCR_014565 | |
| Software, algorithm | PLUMED2.3.0 | Ref. 56 | RRID:SCR_021952 | |
| Software, algorithm | BME | Ref. 58 | RRID:SCR_021953 | |
| Software, algorithm | CcpNmr Analysis | Ref. 64 | RRID:SCR_016984 | |
| Software, algorithm | FLYA | Ref. 65 | RRID:SCR_014229 | |

## Materials

All chemicals were from Sigma-Aldrich unless otherwise stated. DDM was from Carbosynth (UK), match-out-deuterated DDM (sDDM *Midtgaard et al., 2018*) and match-out deuterated POPC (d-POPC) were synthesised at the National Deuteration Facility at ANSTO (Lucas Heights, Australia). The d-POPC was synthesised as previously (*Yepuri et al., 2016*), but with custom deuteration (94% D in tail groups, 71 %D in head group). Details of the synthesis and chemical and isotopic analysis are described in Appendix 1.

## Protein production and purification

For SANS, CorA was produced and purified essentially as described elsewhere (*Johansen et al., 2019*). For studies in DDM, the N-terminal His-tag was cleaved by tobacco etch virus (TEV) protease before gel filtration, whereas for nanodiscs, it was cleaved after incorporation (see below). For NMR, uniformly isotopically labelled CorA was produced in M9 medium containing 3 g/L $^{13}$C-glucose and 1 g/L $^{15}$N-ammonium chloride. $^{15}$N-alanine-labeled CorA was produced in M9 medium containing regular $^{12}$C-glucose and $^{14}$N-ammonium chloride and supplemented with 200 mg/L of $^{15}$N-alanine. Match-out deuterated circularised membrane scaffold protein (MSP), d-csMSP1E3D1, was produced at the D-lab at ILL (Grenoble, France) and purified as described previously (*Johansen et al., 2019*). Proteins were stored at –80 °C until used.

## CorA reconstitution in sND for SANS

CorA, d-csMSP1E3D1, and d-POPC solubilized in cholate were mixed in a ratio of 1:4:400 with a final d-POPC concentration of 10 mM. Cholate and DDM were removed by adding 50 % w/v amberlite XAD-2 overnight at 5 °C. CorA-loaded sNDs were purified by IMAC on NiNTA resin, added TEV protease and dialysed for three hours at RT followed by ON dialysis at 4 °C against 20 mM TrisHCl pH 8, 100 mM NaCl, 0.5 mM EDTA, 1 mM DTT. Finally, the cleaved sample was purified by reverse IMAC on NiNTA resin. The sample was concentrated to approximately 30 µM before the SANS experiment.

## Reconstitution into multilamellar vesicles for NMR

CorA was reconstituted into DMPC at a lipid-to-protein ratio of 0.5 (w/w) by dialysis against 10 mM Tris-HCl pH 8, 40 mM MgCl$_2$, 1 mM methyl-β-cyclodextrin using a 25 kDa MWCO membrane at RT. A white precipitate of multilamellar vesicles was visible after 48 hr and collected by centrifugation after

72 hr. Samples were packed into 1.3 mm or 0.7 mm MAS NMR rotors using an ultracentrifuge tool (Giotto Biotech) at 100,000 g for 1 hr.

## SEC-SANS

Samples were measured at the D22 beamline at ILL, using the SEC-SANS mode described elsewhere (*Johansen et al., 2018*; *Jordan et al., 2016*), but with the upgrade that UV absorption was measured on the flow cell in the same place, but perpendicular to the neutron beam. The setup was placed in a temperature-controlled cabinet at 11 °C. CorA in sDDM was run on a Superdex200 Increase 10/300 GL column (GE Healthcare) in 20 mM TrisDCl pH 7.5, 150 mM NaCl, 1 mM DTT, 0.5 mM sDDM in $D_2O$ with an initial flow rate of 0.3 ml/min to allow full exchange into sDDM (*Midtgaard et al., 2018*). CorA in sND was run on a Superose6 Increase 10/300 GL column (GE Healthcare) in 20 mM TrisDCl pH 7.5, 150 mM NaCl, 1 mM DTT in $D_2O$ with an initial flow rate of 0.5 ml/min. For all samples, the flow was reduced to 0.05 ml/min during elution of the main peak. Measurements were run twice to obtain data from two sample-to-detector distances, 2 m and 11 m, yielding data in a *q*-range of 0.0044 Å$^{-1}$ to 0.42 Å$^{-1}$ with the neutron wavelength of 6 Å. The scattering intensity, *I(q)*, was brought to absolute scale in units of cm$^{-1}$ by normalizing to the direct beam intensity measured with an attenuator in place. All SANS data and metafiles are from ILL-DATA.8-03-940 as in https://doi.ill.fr/105291/ILL-DATA8-03-940.

Pair distance, *p(r)*, distributions were calculated by Bayesian indirect Fourier transformation at the bayesapp server (*Hansen, 2014*) (available from the genapp server at https://somo.chem.utk.edu/bayesapp/). *p(r)*s and the corresponding *I(q)*s for PDB structures were calculated using CaPP (available at github.com/Niels-Bohr-Institute-XNS-StructBiophys/CaPP, copy archived at swh:1:rev:210c-958f0aa476a2194871ced15513c5545da8aa, *Larsen, 2022*) with a water layer excess density of 6% applied to the parts of the protein outside the membrane.

## Model building

For use in both SANS comparison, normal mode analysis, and simulations, we rebuilt the missing residues and sidechains of the closed PDB structure 4I0U. The 2–6 missing residues at each chain terminus was rebuilt using Modeller's automodel functionality (*Webb and Sali, 2017*). In the open cryo-EM structures, between 18 and 20 residues were missing in the N-terminus. These were rebuilt from the closed X-ray structure (4I0U) to allow for direct comparison to SANS measurements.

## Normal mode analysis

Non-linear normal mode analysis was performed using the NOn-Linear rigid Block NMA (*Hoffmann and Grudinin, 2017*) (NOLB) routine (NOLB, RRID:SCR_021954). The NOLB routine was integrated with PEPSI-SANS 2.2 (PESI-SANS, RRID:SCR_021950) and a regularization algorithm to avoid unphysical structures. In the latter, $T = \chi^2 + \alpha S$ is minimized, where $S$ constraints the secondary structure deviation. Scanning over different values of $\alpha$, a plot of $\chi^2$ vs. $S$ is obtained. The best compromise between conservation of structure and the best fit to the data is chosen from the 'elbow'-region (*Figure 4—figure supplement 2*).

## Metadynamics and reweighting

The simulation system was set up using the MARTINI3.0b coarse grained force field (Martini3, RRID:SCR_021951) (*Marrink, 2021*) with elastic networks terms applied to the individual chains only with a 0.9nm cut-off and a force constant of 500kJ/(mol nm$^2$). As MARTINI does not contain parameters for protein-bound $Mg^{2+}$ but rather models it as a free ligand with four waters bound, the $Mg^{2+}$ ions bound in the structure were deleted. To avoid overly electrostatic repulsions from the remaining $Mg^{2+}$ coordinating amino acids (Asp89, Asp179, Asp253 and Glu88), they were transformed into their protonated states. The POPC membrane was obtained from CHARMM-GUI (*Jo et al., 2009*; *Qi et al., 2015*) using 450 POPCs in each bilayer, ensuring that the entire system was big enough for larger conformational changes. The system was solvated with the MARTINI non-polarizable water and neutralized with 300mM NaCl comparable to the SANS experimental set-up. The equilibration was performed according to the CHARMM-GUI equilibration protocol using a minimization step followed by six equilibration steps with slow decrease in the positional restraint forces on both lipids and protein in each step (*Qi et al., 2015*).

The GROMACS-5.1.4 software package (GROMACS, RRID:SCR_014565) (*Abraham et al., 2015*) was used to simulate with a 20 fs time step. Temperature and semi-isotropic pressure were controlled at 303.15K and 1bar using the stochastic velocity rescaling thermostat (*Bussi et al., 2007*) and Parrinello-Rahman barostat (*Parrinello and Rahman, 1981*). Electrostatic interactions were treated using the reaction field approach. The cutoff of short-range distance for the electrostatic interactions was 1.1nm. The potential shift Verlet scheme was used to cut off the Lennard-Jones potential at long ranges.

Well-Tempered Metadynamics (*Barducci et al., 2008*) simulations were performed with the PLUMED2.3.0 software (PLUMED 2, RRID:SCR_021952) (*Tribello et al., 2014*). A radius of gyration collective variable was applied on all backbone beads for the intracellular residues 170-190 and 220-250 of all five chains. The metadynamics parameters were set as follows: Gaussian width = 0.05, Gaussian height = 2.1, Gaussian deposition stride = 100, biasfactor = 15and an upper wall defined at a CV radius of gyration of 4.0nm. The wall was defined as a harmonic restraint with a force constant = 10000, harmonic exponential power = 4, off-set = 0, and a rescaling factor of 1. Multiple wall types and sizes were attempted, with lower walls causing too little dynamics for fitting with the experimental SANS data and higher walls causing individual monomers to bend unphysically and giving an unfeasible large sample space. Clustering of the simulation trajectory was performed using the KMeans clustering method (*Tiberti et al., 2015*).

The software program BME (BME, RRID: SCR_021953) was used to reweight the MetaD trajectory to fit the experimental SANS data (*Bottaro et al., 2020*). The hyperparameter θ was determined based on a L-curve analysis (analogous to the procedure in normal mode analysis, see *Figure 4— figure supplement 2*) of $S_{rel}$ vs $\chi^2$, where θ is chosen where a natural kink is observed and any further decrease in $\chi^2$ gives rise to an increasing larger penalty in $S_{rel}$. As the simulations are not fully converged and the chosen force field is coarse grained, we set the trust in the force field lower and chose a slightly lower θ -value than the kink observed.

To account for a fixed fraction of symmetric pentamers, a differential intensity was derived by $I_{diff}$ = $I_{exp}$ − f · $I_{calc, 5sym}$, where f is the fraction of symmetric pentamer, $I_{exp}$ is the experimental SANS data, and $I_{calc,5sym}$ is the SANS signal of PDB 4I0U calculated by PEPSI-SANS. The forward scattering of the calculated SANS signal, $I_{calc,5sym}(0)$, was scaled to match the forward scattering of the experimental SANS data. Reweighting was done against the differential SANS signal.

## Solid-state NMR spectroscopy

Spectra of uniformly labelled samples were measured at a magnetic field of either 19.5T or 23.4T corresponding to a $^1H$ Larmor frequency of 800 MHz and 1000 MHz, respectively. The spectrometers were equipped with a Bruker 0.7 mm MAS probe, spinning at 107 kHz, at a constant temperature of 300 K. Spectra of $^{15}N$ Ala-labeled samples were acquired at 23.4T on a Bruker 1.3 mm MAS probe, spinning at 60 kHz, at a constant temperature of 300 K. The assignment of backbone resonances was obtained by acquiring a set of eleven $^1H$-detected 3D experiments as described elsewhere (*Schubeis et al., 2018*; *Lalli et al., 2017*). This set notably involved six amide-detected experiments [(H)CANH, (H)CONH, (H)(CO)CA(CO)NH, (H)CO(CA)NH, (H)(CA)CB(CA)NH, (H)(CA)CB(CACO)NH] (*Barbet-Massin et al., 2014*) and five Hα-detected experiments [(H)NCAHA, (H)N(CO)CAHA, (H)COCAHA, (H)CO(N)CAHA, (H)CBCAHA] (*Stanek et al., 2016*), linking resonances over neighboring residues thanks to common $^{13}C$ or $^{15}N$ shifts. Adamantane was used as the external reference. Spectral analysis and assignment were accomplished with CcpNmr Analysis (CCPN Analysis, RRID:SCR_016984) (*Vranken et al., 2005*) and FLYA (*Schmidt and Güntert, 2012*) available from the CYANA program package (CYANA, RRID:SCR_014229).

Relaxation experiments were based on a $^1H$,$^{15}N$ $^1H$-detected CP-HSQC experiment incorporating an appropriate relaxation delay (*Knight et al., 2012*). 38 and 33 residues spanning different regions of the proteins were used for the uniformly labelled sample in the presence and in the absence of $Mg^{2+}$, respectively. The measurements of site-specific $^{15}N$ $R_{1\rho}$ rates were performed at 107 kHz MAS, 19.5T and 300 K or 280 K, using relaxation delays of 0.05, 1, 5, 15, 50, 100, 200 ms under a spin-lock field of 15 kHz. Measurements of bulk $^{15}N$ $R_1$ were performed at 107 kHz MAS, 19.5T and 300 K, using relaxation delays of 0.5, 1, 2.8, 6.8, 15.8, 23.8, 53.5, 80 s. Measurements of $^{15}N$ $R_{1\rho}$ rates were additionally performed on the $^{15}N$ Ala labelled samples in the presence and in the absence of $Mg^{2+}$ at 60 kHz MAS, 23.5T and 300 K using relaxation delays of 0.1, 1, 5, 10, 25, 50, 100ms under a spin-lock

**Table 1.** Negative stain EM statistics for 3D model refinement.

| Parameter | 1 mM EDTA | 40 mM MgCl2 |
| --- | --- | --- |
| Pixel size, Å | 3.14 | 3.14 |
| Number of micrographs | 436 | 440 |
| Number of picked particles | 193,606 | 185,577 |
| Final number of particles | 36,176 | 46,544 |
| Resolution, Å | ≈ 15 | ≈ 15 |

field of 15 kHz. The relaxation rates were obtained by fitting the experimental decay curves with a mono-exponential function. The error was estimated from the experimental noise by use of a Monte-Carlo evaluation.

## Activity assay

Large unilamellar vesicles of POPC were prepared by dissolving a lipid film in 10 mM MOPS-KOH pH 7.2, 150 mM KCl, 100 µM EGTA including 10 µM Mag-Fluo-4 (Thermo) to a POPC concentration of 15 mg/ml, which was extruded through 0.2 µm membrane filters for 35 times using a mini-extruder (Avanti Polar Lipids). CorA was inserted by mixing a sample of 10.5 mg/ml LUVs, 2 µM CorA, 10 µM Mag-Fluo-4 and 50 mM octyl glucoside. Biobeads SM-2 were added to 45 % w/v and incubated at RT for 30 min, before purifying and at the same time exchanging the extravesicular buffer to measurement buffer (10 mM MOPS-KOH pH 7.2, 150 N-methyl-D-glucamine-HCl, 100 µM EGTA) on Sephadex G50 resin. 20 µl of CorA-LUVs (or plain LUVs) were diluted to a total of 1 ml in measurement buffer (prepared in $H_2O$ or $D_2O$, respectively) containing 10 µM valinomycin with or without 1 mM Co[NH$_3$]$^{3+}$ present. CorA activity was monitored by Mag-Fluo-4 fluorescence at 515 nm (excitation at 488 nm) on a FluoroMax fluorometer (Horiba) upon addition of 10 mM $MgCl_2$ (from a 1 M stock prepared in $H_2O$ or $D_2O$, respectively). The signal was normalized to the flat signal recorded before addition of $MgCl_2$.

## Negative stain EM

CorA was purified by SEC and diluted to 0.1 µM in appropriate buffers containing 1 mM EDTA or 40 mM $Mg^{2+}$. Copper grids were neutralized with an Easiglow glow discharger (Agar Scientific). 3 µl of sample was applied to the grid and incubated for 30 s. The grid was blotted onto a filter paper from the edge, and 3 µL of 2% uranyl formate was added immediately and incubated for 30 s. The staining procedure was repeated two more times. After the final staining, the grid was left to dry for ten minutes. EM data were acquired on a Tecnai TEM (FEI, Thermo Fischer scientific) at Aarhus University, Denmark. The micrographs were processed by XMIPP to *.mcp files, and particle picking, 2D class averages and 3D model refinement was done in Relion 3.0. Statistics for the 3D refinement are given in *Table 1*.

Molecular graphics were performed with UCSF Chimera, developed by the Resource for Biocomputing, Visualization, and Informatics at the University of California, San Francisco, with support from NIH P41-GM103311 (*Pettersen et al., 2004*).

## Acknowledgements

We thank Elliot Gilbert for his assistance with SANS experiments at QUOKKA at ANSTO and Marta Brennich for her assistance with SAXS experiments at BM29 at the ESRF. Thomas Boesen is acknowledged for his help with EM experiments conducted at Aarhus University and Michael Gajhede for his assistance in EM data processing. We thank Ida Louise Jørgensen for helping with functional reconstitution of CorA in large unilamellar vesicles and Michael Maguire for providing a plasmid encoding CorA.

## Additional information

### Funding

| Funder | Grant reference number | Author |
|---|---|---|
| Lundbeckfonden | R155-2015-2666 | Kresten Lindorff-Larsen Lise Arleth |
| Novo Nordisk Fonden | NNF15OC0016670 | Lise Arleth |
| Biotechnology and Biological Sciences Research Council | BB/R00126X/1 | Mark SP Sansom |
| Biotechnology and Biological Sciences Research Council | BB/N000145/1 | Mark SP Sansom |
| Engineering and Physical Sciences Research Council | EP/R004722/1 | Mark SP Sansom |
| Engineering and Physical Sciences Research Council | EP/R029407/1 | Mark SP Sansom |
| Engineering and Physical Sciences Research Council | EP/V010948/1 | Mark SP Sansom |
| Wellcome Trust | 208361/Z/17/Z | Mark SP Sansom |
| National Collaborative Research Infrastructure Strategy | | Tamim Darwish |
| European Commission | INFRAIA-01-2018-2019 GA 871037 (iNext Discovery) | Tobias Schubeis Guido Pintacuda |
| Villum Fonden | 35955 | Nicolai Tidemand Johansen |
| Horizon 2020 - Research and Innovation Framework Programme | ERC-2015-CoG GA 648974 | Guido Pintacuda |

The funders had no role in study design, data collection and interpretation, or the decision to submit the work for publication.

### Author contributions

Nicolai Tidemand Johansen, Conceptualization, Formal analysis, Investigation, Project administration, Validation, Visualization, Writing – original draft, Writing – review and editing; Marta Bonaccorsi, Formal analysis, Investigation, Methodology, Validation, Visualization, Writing – original draft, Writing – review and editing; Tone Bengtsen, Data curation, Formal analysis, Methodology, Software, Visualization, Writing – review and editing; Andreas Haahr Larsen, Data curation, Formal analysis, Investigation, Resources, Software, Visualization, Writing – review and editing; Frederik Grønbæk Tidemand, Pie Huda, Investigation, Resources, Writing – review and editing; Martin Cramer Pedersen, Formal analysis, Investigation, Supervision, Visualization, Writing – original draft; Jens Berndtsson, Resources, Writing – review and editing; Tamim Darwish, Funding acquisition, Project administration, Writing – review and editing; Nageshewar Rao Yepuri, Resources, Visualization, Writing – review and editing; Anne Martel, Data curation, Resources, Writing – review and editing; Thomas Günther Pomorski, Mikaela Rapp, Supervision, Writing – review and editing; Andrea Bertarello, Formal analysis, Investigation, Writing – review and editing; Mark Sansom, Funding acquisition, Supervision, Writing – review and editing; Ramon Crehuet, Methodology, Software, Writing – review and editing; Tobias Schubeis, Conceptualization, Formal analysis, Investigation, Supervision, Visualization, Writing – original draft, Writing – review and editing; Kresten Lindorff-Larsen, Guido Pintacuda, Lise Arleth, Conceptualization, Funding acquisition, Supervision, Writing – original draft, Writing – review and editing, Investigation

### Author ORCIDs
Nicolai Tidemand Johansen http://orcid.org/0000-0002-8596-548X

Marta Bonaccorsi http://orcid.org/0000-0001-6177-0701
Tone Bengtsen http://orcid.org/0000-0001-8423-2156
Andreas Haahr Larsen http://orcid.org/0000-0002-2230-2654
Frederik Grønbæk Tidemand http://orcid.org/0000-0001-8914-9626
Martin Cramer Pedersen http://orcid.org/0000-0002-8982-7615
Pie Huda http://orcid.org/0000-0002-2197-4993
Jens Berndtsson http://orcid.org/0000-0001-6627-8134
Tamim Darwish http://orcid.org/0000-0001-7704-1837
Nageshewar Rao Yepuri http://orcid.org/0000-0002-4665-1306
Thomas Günther Pomorski http://orcid.org/0000-0002-4889-0829
Andrea Bertarello http://orcid.org/0000-0003-3705-1760
Mark Sansom http://orcid.org/0000-0001-6360-7959
Mikaela Rapp http://orcid.org/0000-0002-4401-9518
Ramon Crehuet http://orcid.org/0000-0002-6687-382X
Tobias Schubeis http://orcid.org/0000-0003-2203-1126
Kresten Lindorff-Larsen http://orcid.org/0000-0002-4750-6039
Guido Pintacuda http://orcid.org/0000-0001-7757-2144
Lise Arleth http://orcid.org/0000-0002-4694-4299

**Decision letter and Author response**
Decision letter https://doi.org/10.7554/eLife.71887.sa1
Author response https://doi.org/10.7554/eLife.71887.sa2

## Additional files

### Supplementary files
• Transparent reporting form

### Data availability
SANS data have been deposited in SASBDB under IDs SASDM42, SASDM52, SASDM62, SASDM72. EM data have been uploaded to the Electron Microscopy Data Bank under IDs EMD-13326 and EMD-13327. Activity (fluorescence) data have been uploaded to GitHub at https://github.com/Niels-Bohr-Institute-XNS-StructBiophys/CorAData (copy archived at swh:1:rev:94c6ed6e-fa9166781a36307eb7c6d05e125ffa2d). The Metadynamics simulations have been uploaded to GitHub at https://github.com/KULL-Centre/papers/tree/main/2021/CorA-Johansen-et-al (swh:1:rev:511f01bdf1c2b57d68654deee7f15ce019f65d7a). NMR data have been deposited in Biological Magnetic Resonance Data Bank under ID 50959.

The following datasets were generated:

| Author(s) | Year | Dataset title | Dataset URL | Database and Identifier |
|---|---|---|---|---|
| Johansen NT, Pedersen MC, Tidemand Arleth L | 2021 | Cobalt/magnesium transport protein CorA in matched-out deuterated dodecylmaltoside (dDDM) micelles without Mg2+ | https://www.sasbdb.org/data/SASDM42 | SASBDB, SASDM42 |
| Johansen NT, Pedersen MC, Tidemand Arleth L | 2021 | Cobalt/magnesium transport protein CorA in matched-out deuterated dodecylmaltoside (dDDM) micelles with bound Mg2+ | https://www.sasbdb.org/data/SASDM52 | SASBDB, SASDM52 |
| Johansen NT, Pedersen MC, Tidemand Arleth L | 2021 | Cobalt/magnesium transport protein CorA in matched-out deuterated nanodiscs without Mg2+ | https://www.sasbdb.org/data/SASDM762 | SASBDB, SASDM62 |

*Continued on next page*

*Continued*

| Author(s) | Year | Dataset title | Dataset URL | Database and Identifier |
|---|---|---|---|---|
| Johansen NT, Pedersen MC, Tidemand Arleth L | 2021 | Cobalt/magnesium transport protein CorA in match-out deuterated nanodiscs with bound Mg2+ | https://www.sasbdb.org/data/SASDM72 | SASBDB, SASDM72 |
| Schubeis T, Bonaccorsi M, Pintacuda G | 2021 | Resonance assignment of Mg-bound CorA in DMPC | https://bmrb.io/data_library/summary/?bmrbId=50959 | BMRB, ID50959 |
| Larsen AH, Johansen NT, Arleth L | 2021 | T. maritima CorA in DDM micelles without Mg2+ bound in D2O | https://www.ebi.ac.uk/emdb/entry/EMD-13326 | EMDB, EMD-13326 |
| Larsen AH, Johansen NT, Arleth L | 2021 | T. maritima CorA in DDM micelles with Mg2+ bound in D2O | https://www.ebi.ac.uk/emdb/entry/EMD-13327 | EMDB, EMD-13327 |

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

## Appendix 1

### Synthesis of match-out deuterated POPC

The overall synthesis of POPC-$d_{77}$ is reported elsewhere (*Yepuri et al., 2016*). *Appendix 1—figure 1* shows the synthetic scheme followed in this study to produce the specific level of deuteration in the head and tail groups of the POPC. The specific level of deuteration in the tail group was achieved by diluting pure heavy water with light water in specific ratios in the Parr reactor when making the deuterated alkyl chains from their fatty acid precursors (*Midtgaard et al., 2018*). The analysis data and spectra of the intermediate compounds and the final compound are shown in *Appendix 1—figures 2–24*. Electrospray ionisation mass spectra (ESI-MS) were recorded on a 4,000 QTrap AB SCIEX Mass Spectrometer. The overall percent deuteration of the molecules was calculated by ER-MS (enhanced resolution – MS) using the isotope distribution analysis of the different isotopologues by analysing the area under each MS peak which corresponds to a defined number of deuterium atoms. The contribution of the carbon-13 (natural abundance) to the value of the area under each [X + 1] MS signal is subtracted based on the relative amount found in the protonated version. In a typical analysis we measure the C-13 natural abundance contribution by running ER-MS of the protonated version (or estimate it by ChemDraw software) and use this value in our calculation using an in-house developed method that subtracts this contribution from each MS signal constituting the isotope distribution. $^1$H NMR (400 MHz), $^{13}$C NMR (100 MHz), $^{31}$P NMR (162 MHz) and $^2$H NMR (61.4 MHz) spectra were recorded on a Bruker 400 MHz spectrometer at 298 K. Chemical shifts, in ppm, were referenced to the residual signal of the corresponding solvent. Deuterium NMR spectroscopy was performed using the probe's lock channel for direct observation.

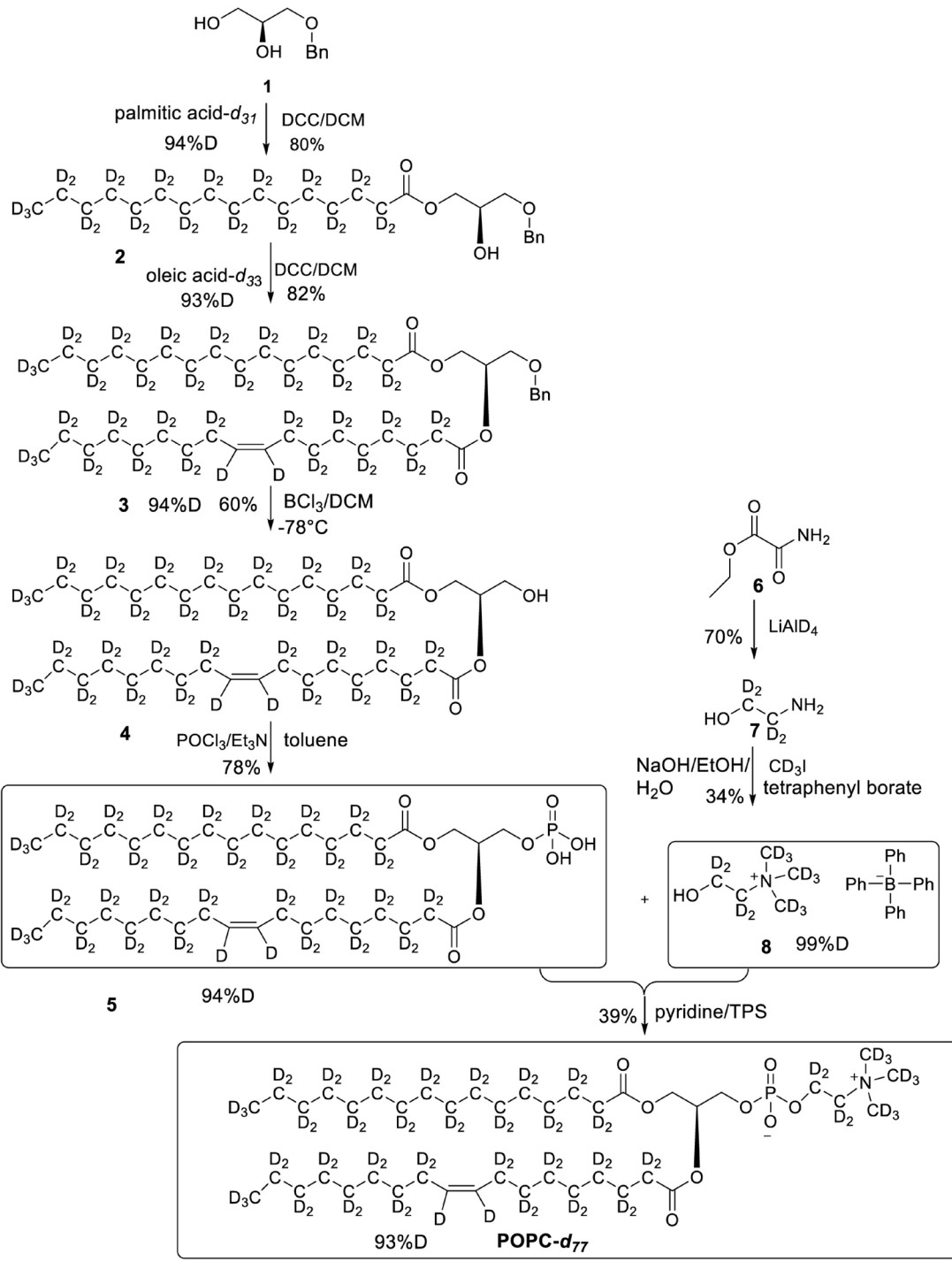

**Appendix 1—figure 1.** Overall synthesis achieved by following a reported paper, all the intermediates and final POPC-d₇₇ were obtained in similar yields (*Yepuri et al., 2016*).

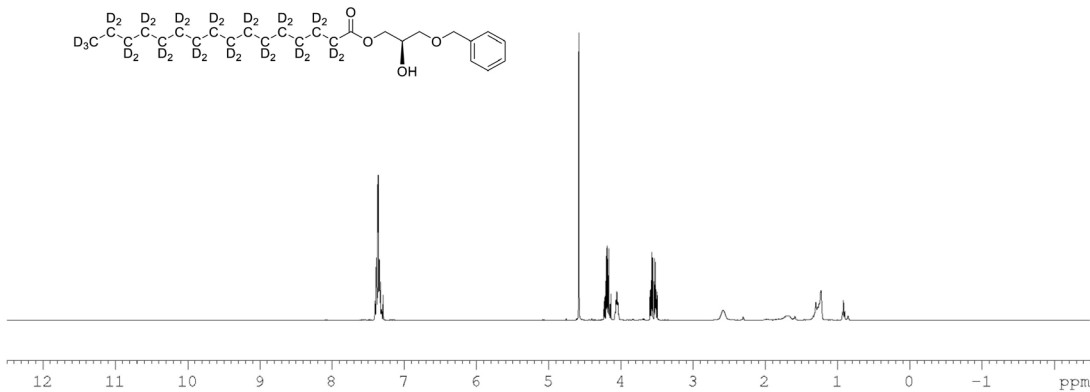

**Appendix 1—figure 2.** [1]H NMR of 1-palmitoyl-d$_{31}$-sn-3-benzyloxy-glycerol (**Appendix 1—figure 1**, molecule **2**) in CDCl$_3$. (400 MHz, CDCl$_3$), δ residual protons 0.88 (m, 0.22 H), 1.10–17 (m, 1.96 H), 1.57.1.86 (m, 1.55 H), 1.96 (m, 0.54 H), 3.69 (m, 2 H), 4.09 (m, 1 H), 4.22 (m, 2 H) 4.60 (s, 2 H), 7.36 (m, 5 H).

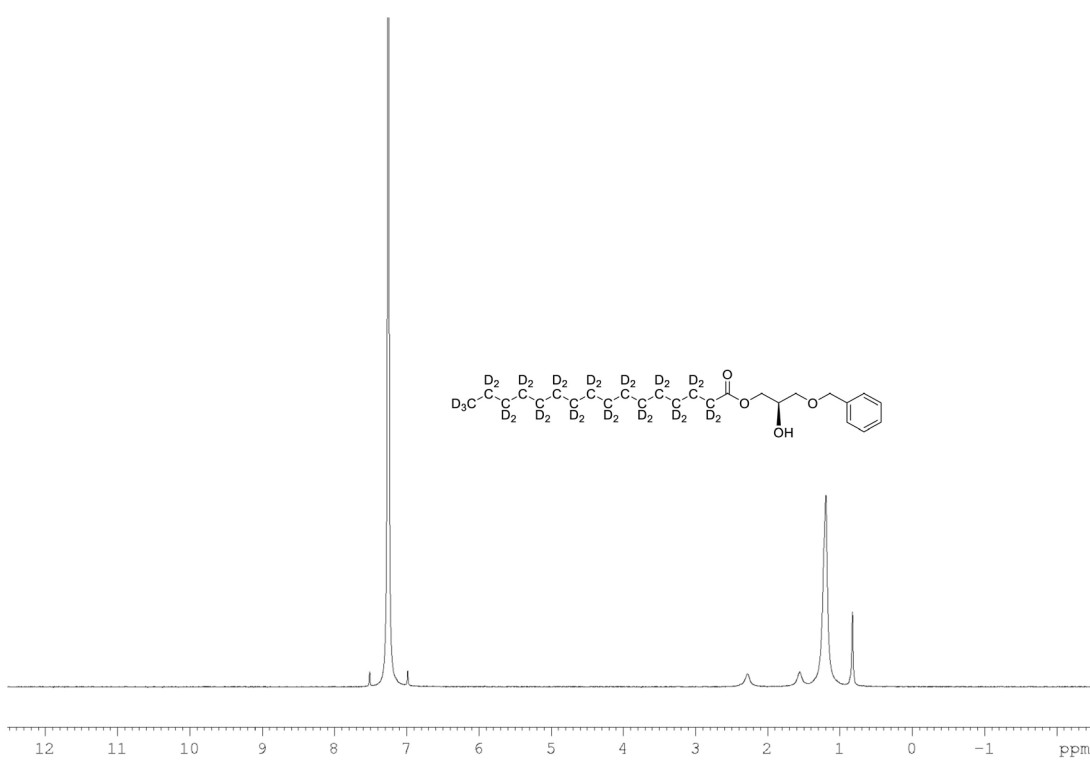

**Appendix 1—figure 3.** [2]H NMR of 1-palmitoyl-d$_{31}$-sn-3-benzyloxy-glycerol (**Appendix 1—figure 1**, molecule **2**) in CDCl$_3$. (400 MHz, CDCl$_3$), δ 0.82 (m), 1.18 (m), 1.54 (m), 2.27 (m).

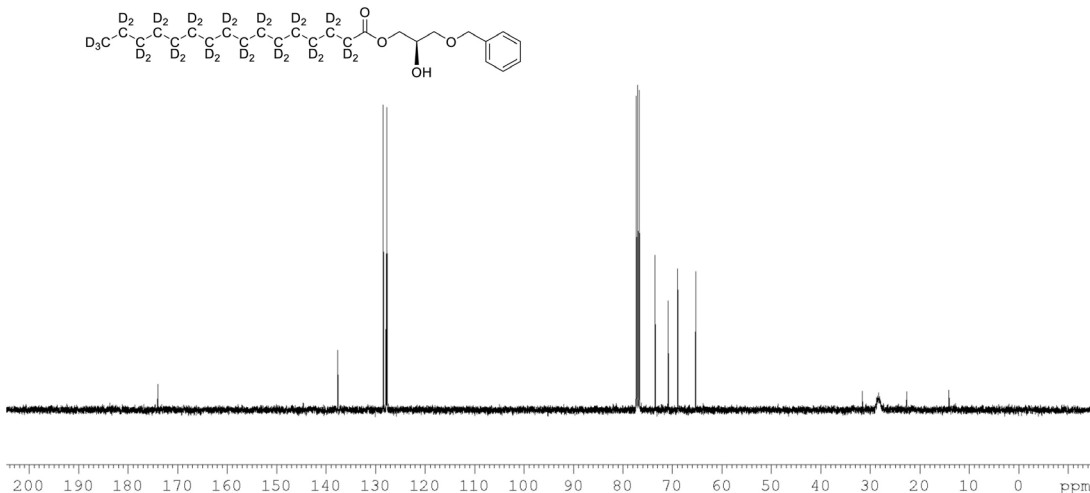

**Appendix 1—figure 4.** $^{13}$C NMR of 1-palmitoyl-d$_{31}$-sn-3-benzyloxy-glycerol (*Appendix 1—figure 1*, molecule **2**) in CDCl$_3$. (400 MHz, CDCl$_3$), 10.9 (m), 22.09 (m), 28.33 (m), 33.05 (m), 65.33, 68.9, 70.80, 73.5, 127.7, 127.9, 128.5, 137.6, 174.0.

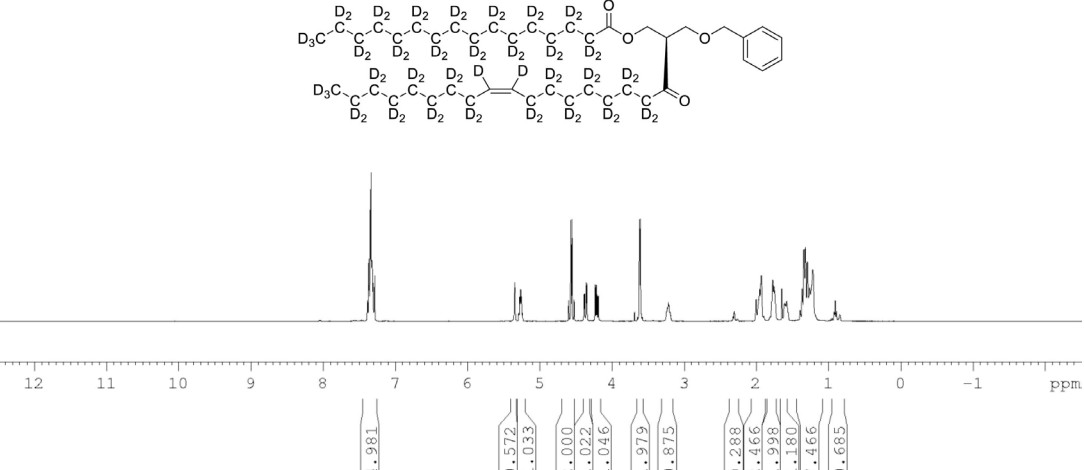

**Appendix 1—figure 5.** $^{1}$H NMR of 1-palmitoyl-d$_{31}$-2-oleoyl-d$_{33}$-sn-3-benzyloxy-glycerol (*Appendix 1—figure 1*, molecule **3**) in CDCl$_3$. (400 MHz, CDCl$_3$), δ residual protons 0.90 (m, 3.56 H), 1.29 (m, 5.15 H), 1.98 (m, 0.68 H), 2.29 (m, 1.25 H), 3.61 (d, J = 5.0 Hz, 2 H), 4.21 (m, 1 H), 4.36 (m, 1 H), protonated benzyl protons 4.56 (AB q, J = 12 Hz, 2 H), 5.26 (m, 1 H), 5.35 (s, 0.65 H), 7.34 (m, 5 H).

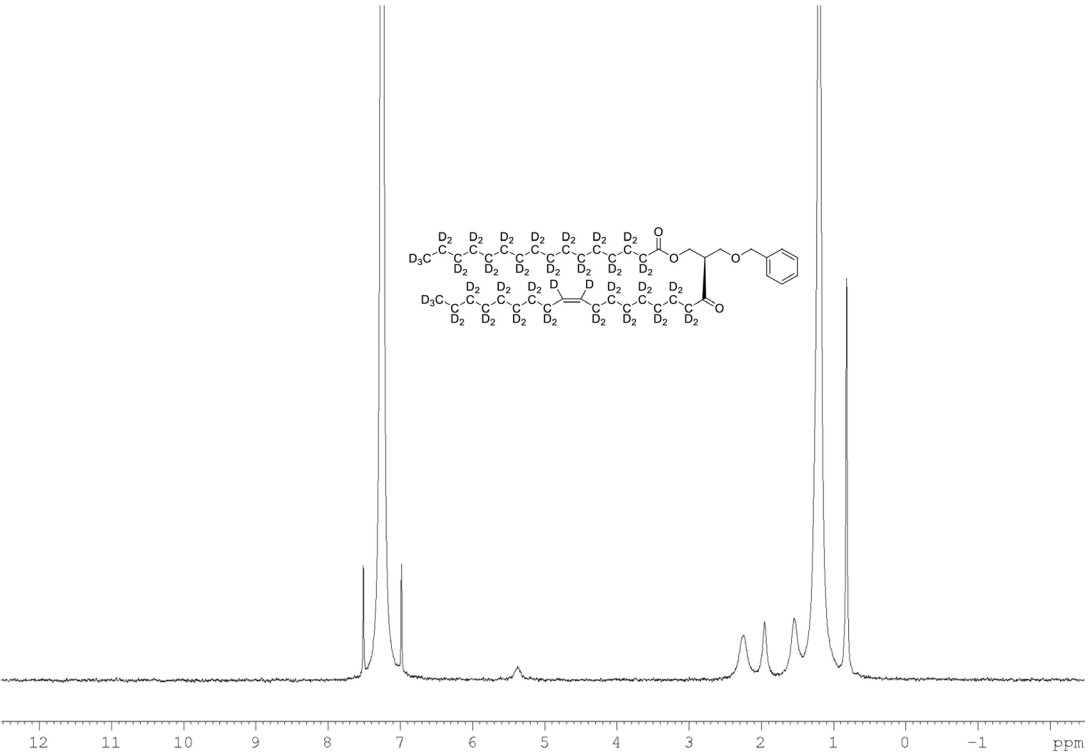

**Appendix 1—figure 6.** $^2$H NMR of 1-palmitoyl-d$_{31}$-2-oleoyl-d$_{33}$-sn-3-benzyloxy-glycerol (*Appendix 1—figure 1*, molecule **3**) in CDCl$_3$. (400 MHz, CDCl$_3$), δ 0.82 (m, 6D), 1.19 (m, 35.35D), 1.53 (m, 3.74D), 1.94 (m, 3.34D), 2.25 (m, 2.63D), 3.56 (m, 1.71D), 4.27 (m, 1.04D), 5.35 (m, 1.92).

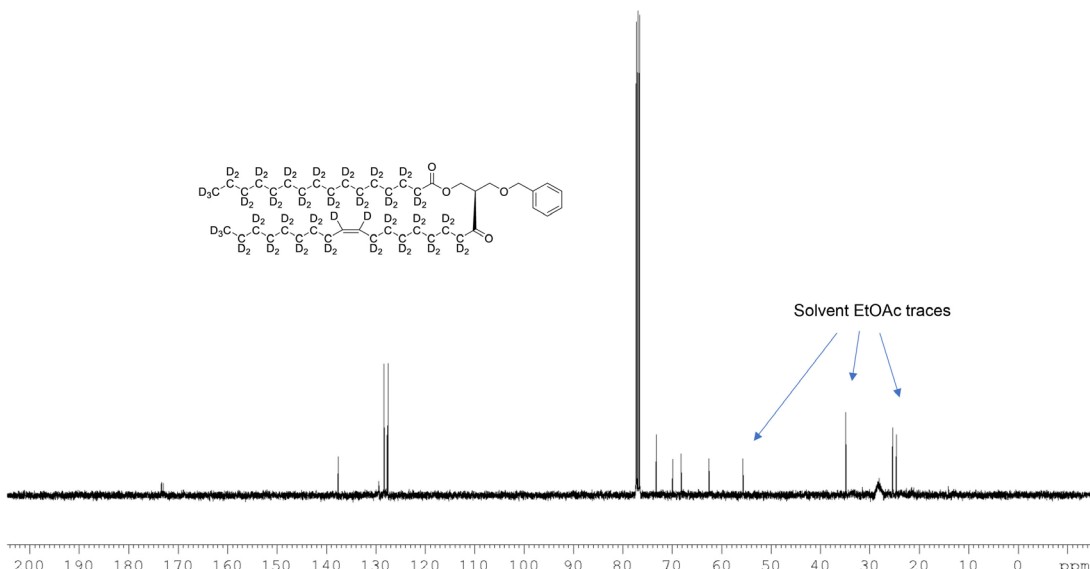

**Appendix 1—figure 7.** $^{13}$C NMR of 1-palmitoyl-d$_{31}$-2-oleoyl-d$_{33}$-sn-3-benzyloxy-glycerol (*Appendix 1—figure 1*, molecule **3**) in CDCl$_3$. (400 MHz, CDCl$_3$), δ 13.08 (m), 21.60 (m), 23.96 (m), 26.50 (m), 28.2 (m), 30.60 (m), 33.80 (m), 62.60, 68.39 9, 70.10, 73.1, 127.7, 127.9, 128.5, 137.7, 173.1, 173.4.

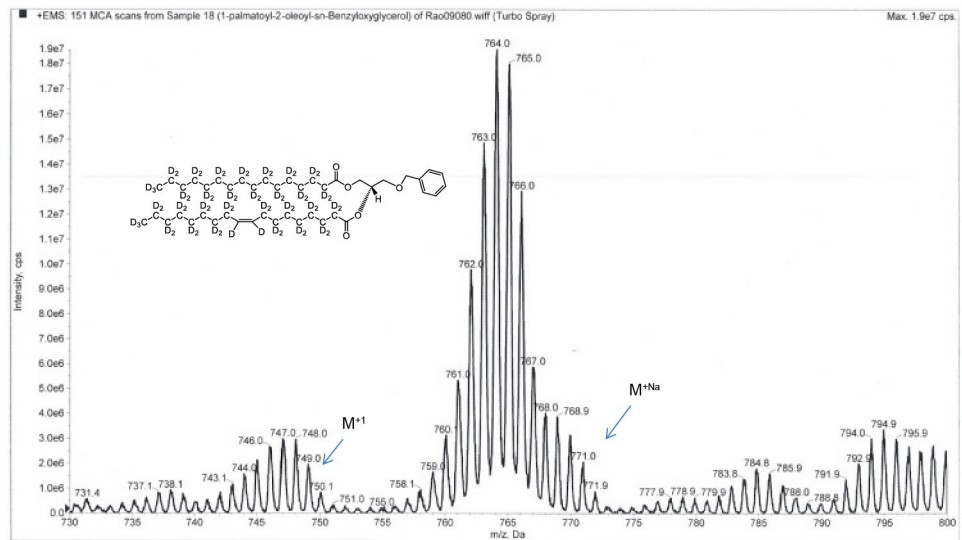

**Appendix 1—figure 8.** ESI-MS, m/z 749 [M$^{+1}$]$^+$ of POPC precursor 1-palmtoyl-2-oleoyl-sn-benzyloxyglycerol-d$_{64}$ (*Appendix 1—figure 1*, molecule **3**). Overall 94%D, isotope distribution d$_{64}$, 7.8%, d$_{63}$, 15.1%, d$_{62}$, 19.5%, d$_{61}$, 18.0%, d$_{60}$, 13.1%, d$_{59}$, 10.2%, d$_{58}$, 6.5%, d$_{57}$, 3.9%, d$_{56}$, 2.7%.

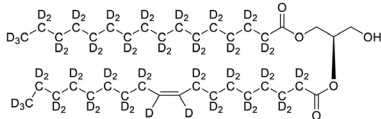

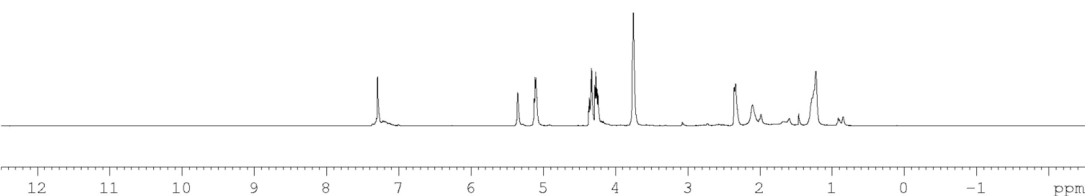

**Appendix 1—figure 9.** $^1$H NMR of 1-palmitoyl-d$_{31}$-2-oleoyl-d$_{33}$-sn-3-glycerol (*Appendix 1—figure 1*, molecule **4**) in CDCl$_3$. (400 MHz, CDCl$_3$), δ residual protons 0.24 (m, 0.47 H), 1.28 (m, 1.51 H), 1.67 (m, 2.24 H), 2.13 (m, 0.52 H), 2.23 (m, 1.29 H), 3.72 (m, 2 H), 4.30 (m, 2 H), 5.10 (m, 1 H), 5.34 (s, 0.48 H).

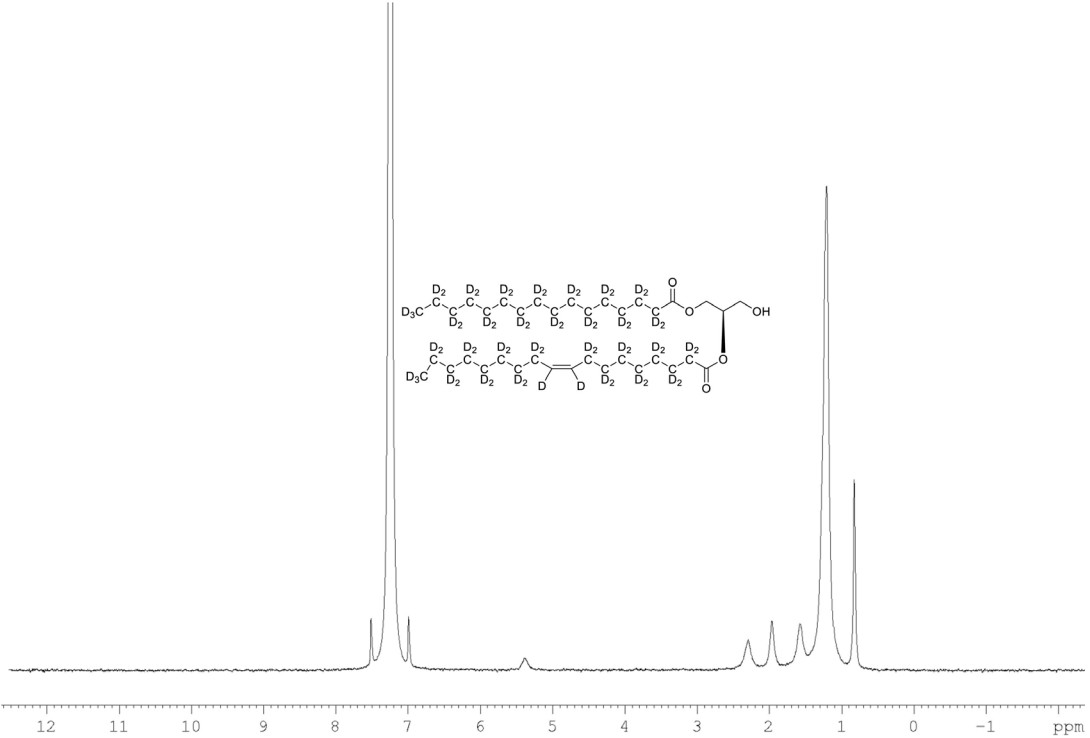

**Appendix 1—figure 10.** $^2$H NMR of 1-palmitoyl-d$_{31}$-2-oleoyl-d$_{33}$-sn-3-glycerol (*Appendix 1—figure 1*, molecule **4**) in CDCl$_3$. (400 MHz, CDCl$_3$), δ 0.82 (m, 8.6D), 1.19 (m, 49.2D), 1.55 (m, 4.8D), 1.94 (m, 3.46D), 2.28 (m, 3.82D), 5.38 (m, 1.1D).

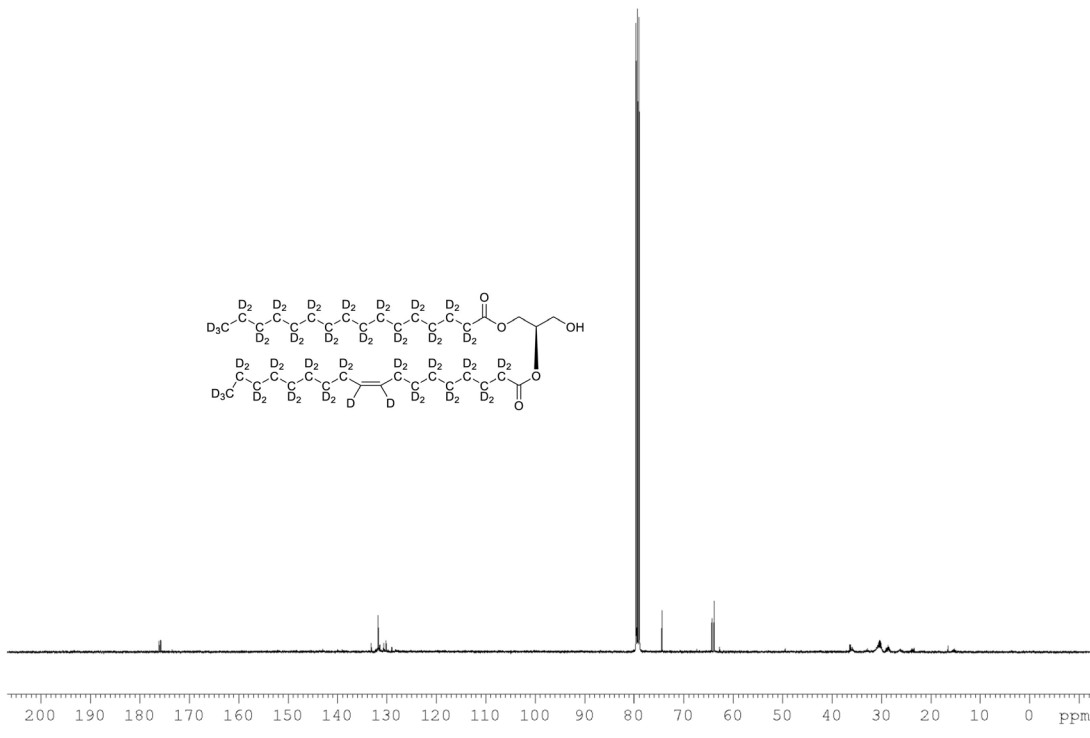

**Appendix 1—figure 11.** $^{13}$C NMR of 1-palmitoyl-d$_{31}$-2-oleoyl-d$_{33}$-sn-3-glycerol (*Appendix 1—figure 1*, molecule **4**) in CDCl$_3$. (400 MHz, CDCl$_3$), δ 12.89 (m), 21.39 (m), 22.6 (m), 24.00 (m), 26.25 (m), 28.2 (m), 30.49 (m), 33.86 (m), 65.00, 60.76, 61.60, 71.64, 128.02, 129.14, 129.5, 173.53, 173.93.

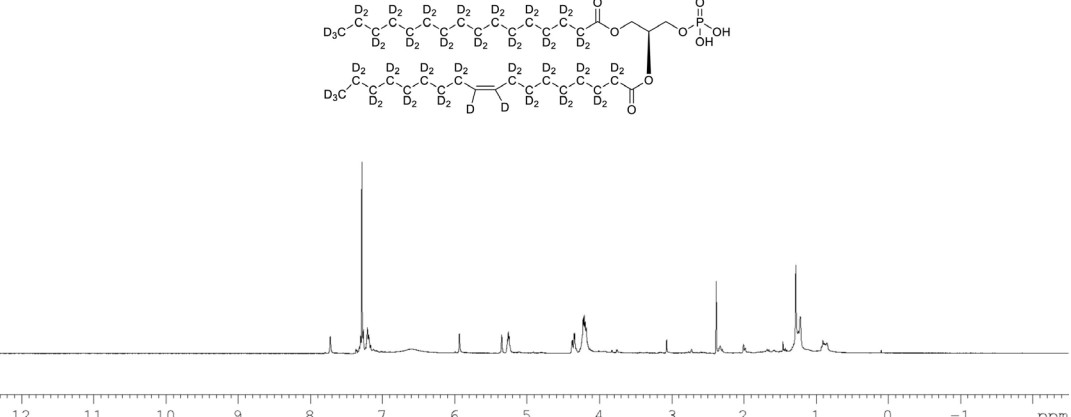

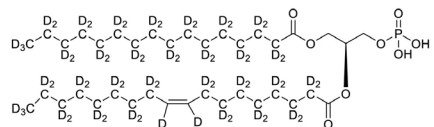

**Appendix 1—figure 12.** $^{1}$H NMR of crude product 1-palmitoyl-d$_{31}$-2-oleoyl-d$_{33}$-sn-3-glycero-phosphatidic acid (*Appendix 1—figure 1*, molecule **5**) in CDCl$_{3}$. This synthesised crude dried product was used in next step without further purification.

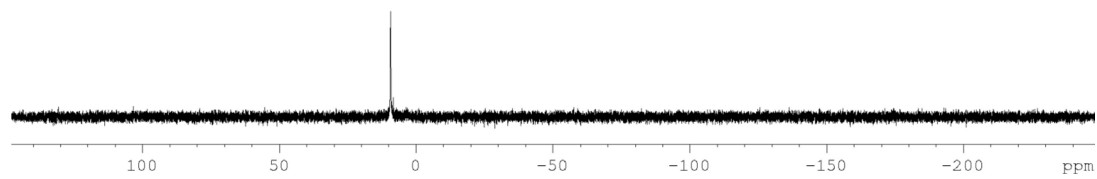

**Appendix 1—figure 13.** $^{31}$P NMR of crude product 1-palmitoyl-d$_{31}$-2-oleoyl-d$_{33}$-sn-3-glycero-phosphatidic acid (*Appendix 1—figure 1*, molecule **5**) in CDCl$_{3}$. This synthesised crude dried product was used in next step without further purification.

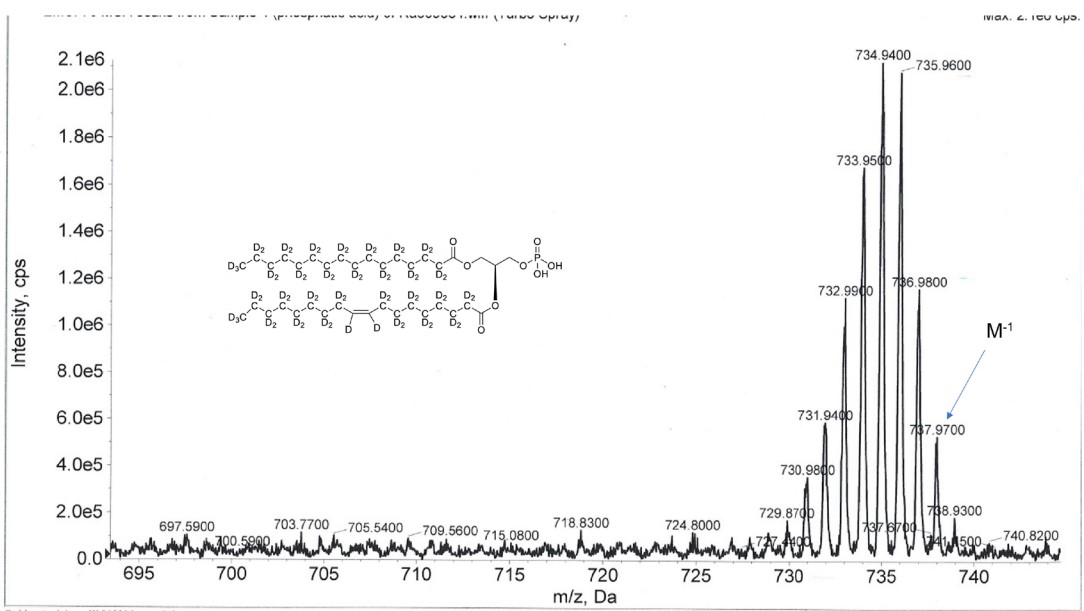

**Appendix 1—figure 14.** ESI-MS, m/z 737 [M$^{-1}$]$^{-}$ of crude product of 1-palmitoyl-d$_{31}$-2-oleoyl-d$_{33}$-sn-3-glycero-phosphatidic acid (*Appendix 1—figure 1*, molecule **5**). Overall 94%D, isotope distribution d$_{64}$, 0.6%, d$_{63}$, 8.9%, d$_{62}$, 18.8%, d$_{61}$, 25.1%, d$_{60}$, 20.6%, d$_{59}$, 13.0%, d$_{58}$, 7.5%, d$_{57}$, 5.2%, d$_{56}$, 0.2%, d$_{55}$, 0.1%.

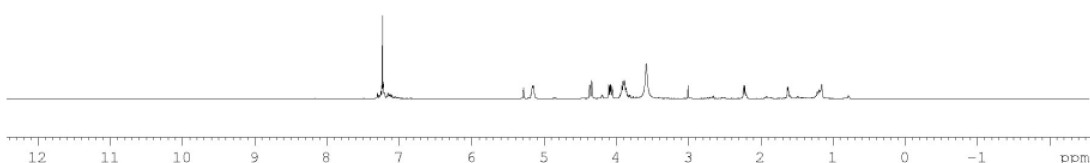

**Appendix 1—figure 15.** $^{1}$H NMR of POPC-d$_{77}$ in CDCl$_{3}$. (400 MHz, CDCl3), δ residual protons 0.85 (m, 0.17 H), 1.25 (m, 1.48 H), 1.54 (m, 0.22 H), 1.97 (m, 0.18 H), 2.22 (m, 0.66 H), 3.89 (m, 2 H), 4.08 (m, 1 H), 4.34 (m, 1 H), 5.14 (m, 1 H), 5.27 (s, 0.43 H).

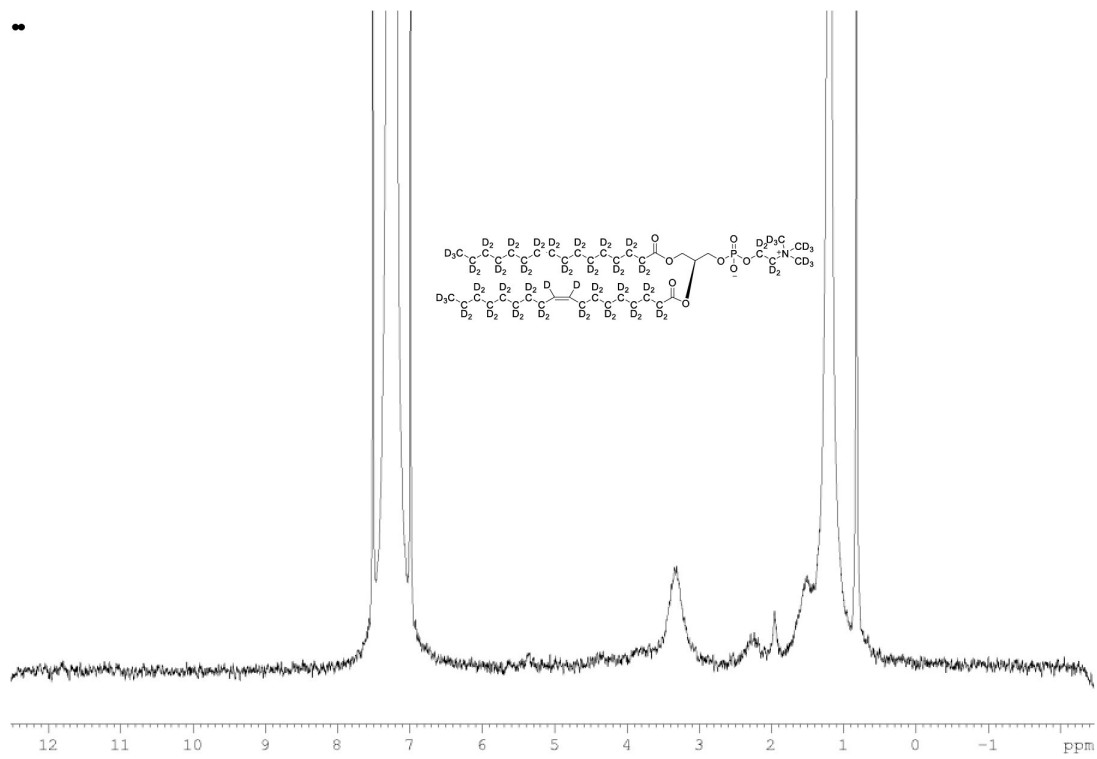

**Appendix 1—figure 16.** $^2$H NMR of POPC-d$_{77}$ in CDCl$_3$. (400 MHz, CDCl$_3$), δ 0.80 (m), 1.19 (m), 2.20 (m), 1.93 (m, 6.0D), 3.35 (m), 3.84 (m), 5.36 (m).

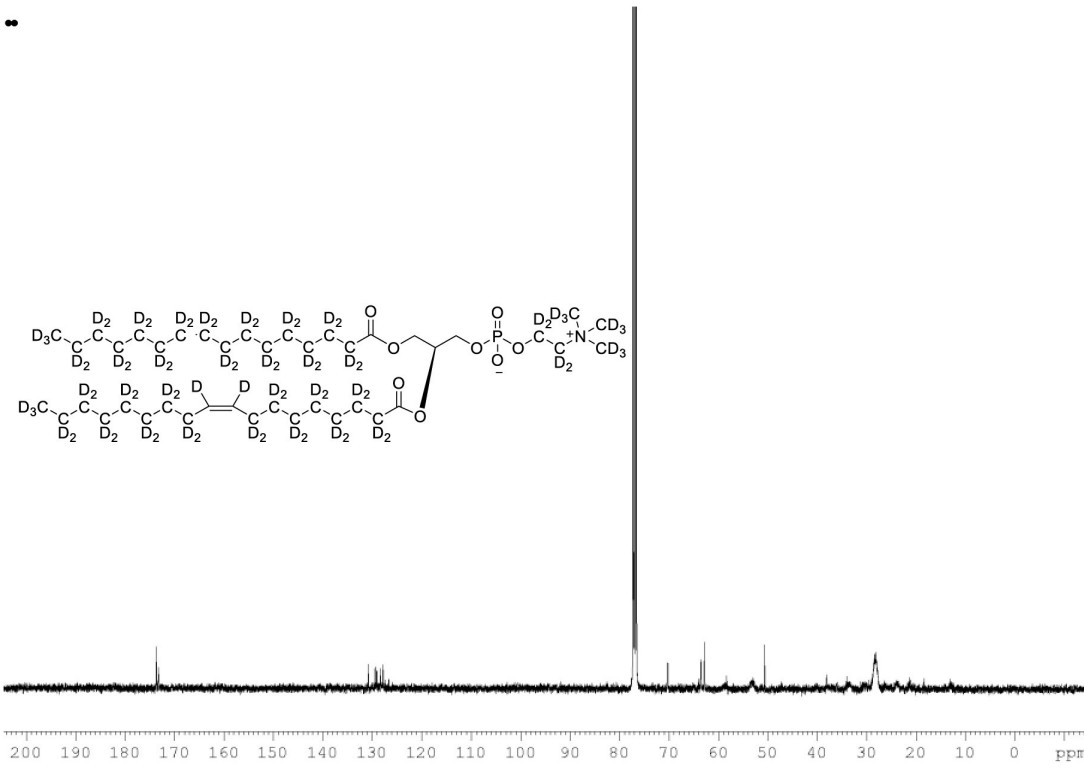

**Appendix 1—figure 17.** $^{13}$C NMR of POPC-d$_{77}$ in CDCl$_3$. (400 MHz, CDCl$_3$), δ 13.03 (m), 21.48 (m), 23.88 (m), 26.24 (m), 28.36 (m), 30.49 (m), 33.72 (m), 53.17, 58.76 (m), 62.24 (m), 69.84 (m), 127.90 (s), 129.5, 173.31, 173.69.

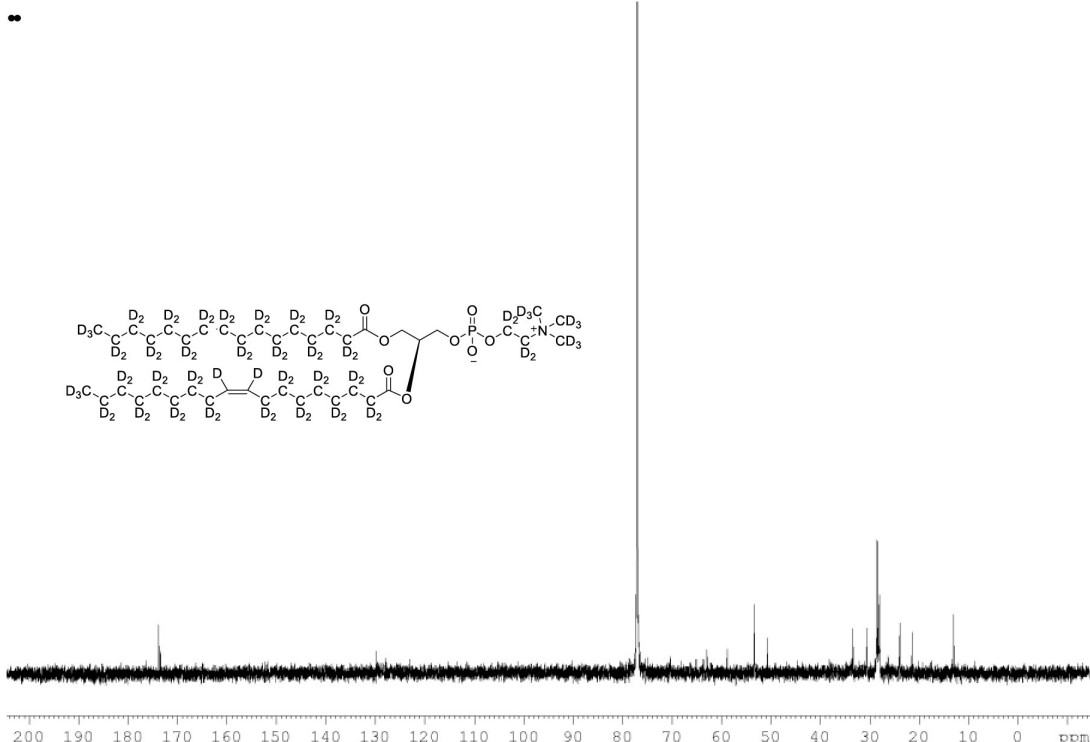

**Appendix 1—figure 18.** {$^1$H}and {$^2$H} decoupled $^{13}$C NMR spectra of POPC-d$_{77}$ in CDCl$_3$.

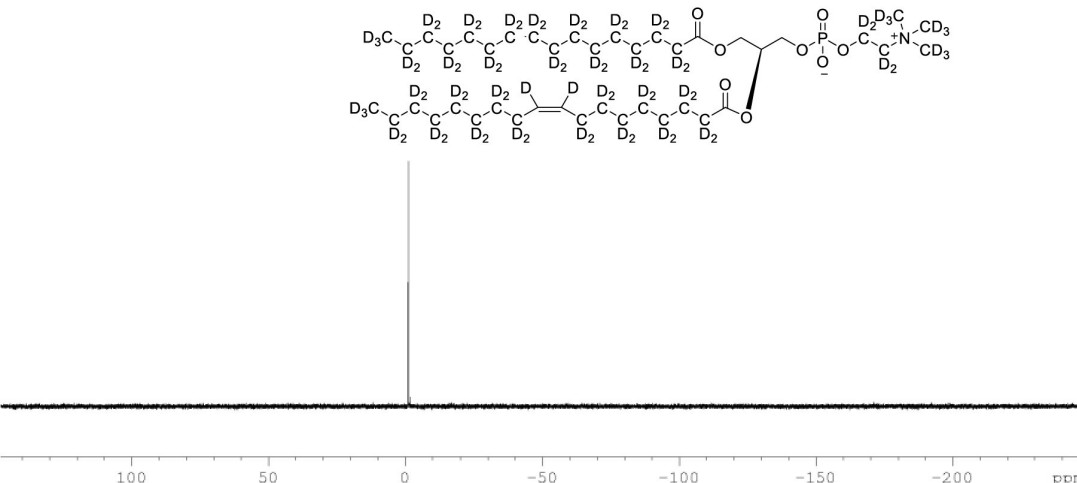

**Appendix 1—figure 19.** $^{31}$P NMR of POPC-d$_{77}$ in CDCl$_3$. (400 MHz, CDCl3), single peak at δ –2.20.

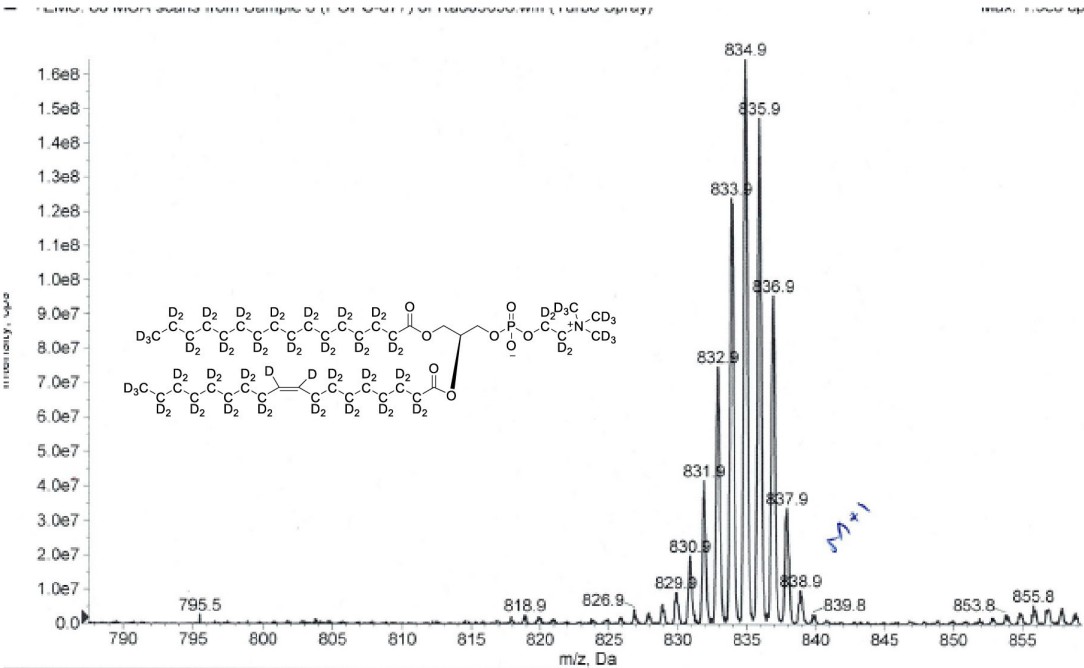

**Appendix 1—figure 20.** ESI-MS, m/z 838 [M$^{+1}$]$^+$ of POPC-d$_{77}$. Overall 93%D, isotope distribution d$_{77}$, 0%, d$_{76}$, 0%, d$_{75}$, 7.5%, d$_{74}$, 21.2%, d$_{73}$, 28.7%, d$_{72}$, 25.3%, d$_{71}$, 15.5%, d$_{70}$, 10.8%.

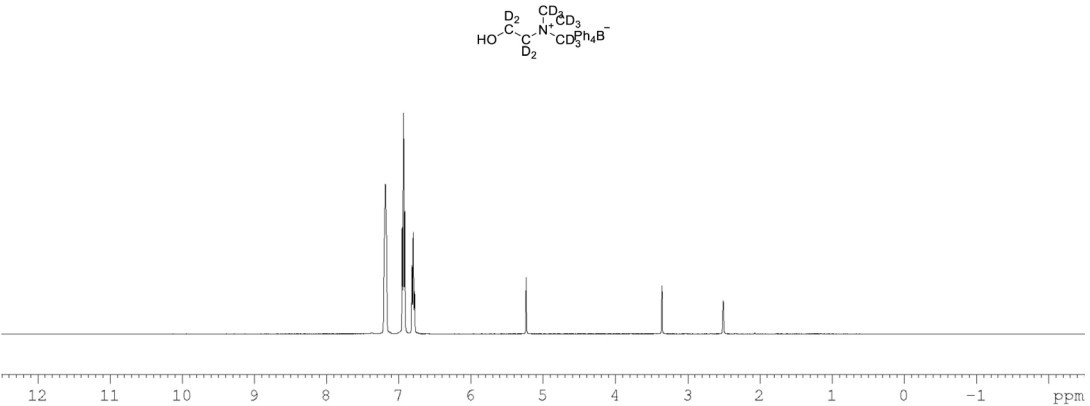

**Appendix 1—figure 21.** $^1$H NMR of choline-d$_{13}$ tetraphylborate (**Appendix 1—figure 1**, molecule **8**) in DMSO-d$_6$. (400 MHz, DMSO-d$_6$) δ 6.85 (m, 4 H), 6.96 (m, 8 H), 7.21 (m, 8 H).

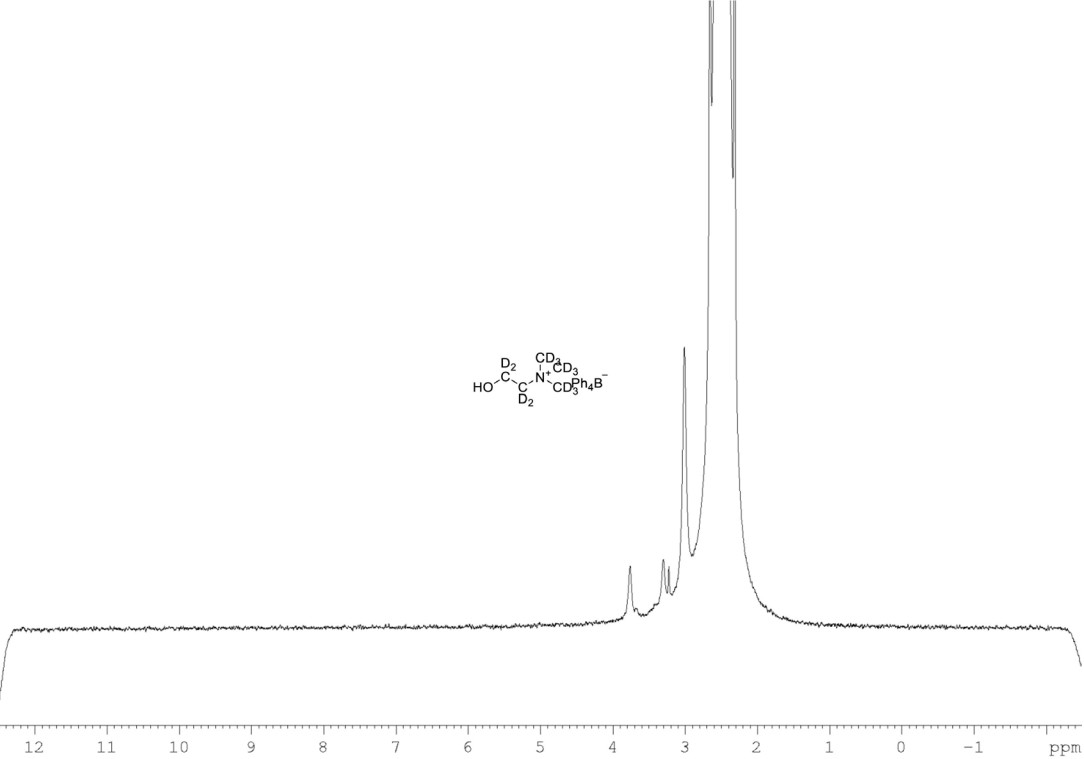

**Appendix 1—figure 22.** ²H NMR of choline-d₁₃ tetraphylborate (*Appendix 1—figure 1*, molecule **8**) in DMSO-d₆. (61.4 MHz, DMSO-d₆) δ 3.30 (m, 9D), 3.32 (m, 2D), 3.78 (m, 2D).

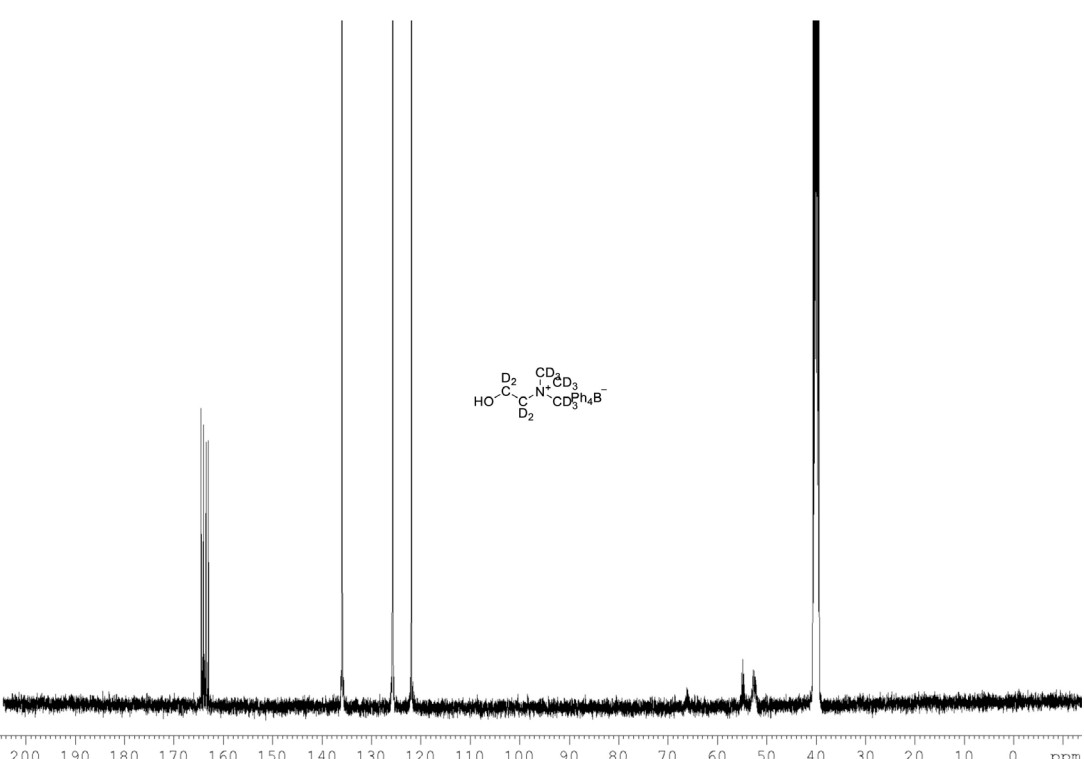

**Appendix 1—figure 23.** ¹³C NMR of choline-d₁₃ tetraphylborate (*Appendix 1—figure 1*, molecule **8**) in DMSO-d₆. (100 MHz, DMSO-d₆) δ 52.5 (m), 54.9 (m), 121.9, 125.9, 136.2, 163.9 (m).

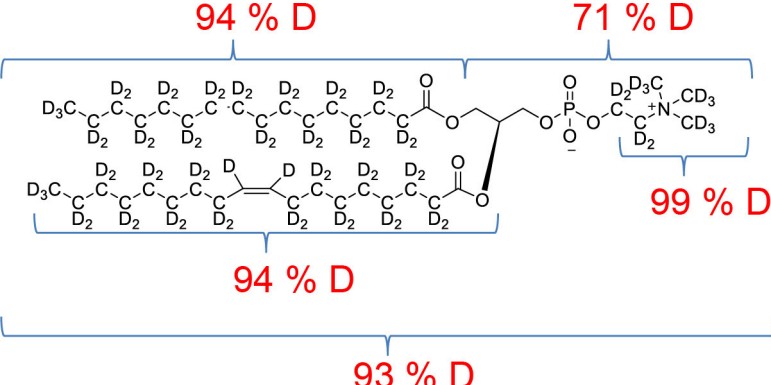

**Appendix 1—figure 24.** POPC-d$_{77}$ percentage distribution of D levels at different sites.

