## [Decision Letter]

**Decision letter after peer review:**

Thank you for submitting your article "Mg^2+^-dependent conformational equilibria in CorA: an integrated view on transport regulation" for consideration by *eLife*. Your article has been reviewed by 2 peer reviewers, and the evaluation has been overseen by a Reviewing Editor and Volker Dötsch as the Senior Editor. The reviewers have opted to remain anonymous.

Your manuscript has been reviewed by two experts who find the work of great interest and very carefully done. The study points to the importance of conformational equilibria for Mg^2+^ transport by CorA membrane receptors and will represent a valuable addition to our understanding of how this important ion is transported. Comments such as (1) "This is an elegant study into the mechanisms of Mg^2+^ transport by CorA membrane receptors. On the basis of multiple types of experimental data (SANS, solid-state NMR, and negative stain EM) and molecular dynamics simulations, the authors uncovered the structural and dynamic basis for the "symmetry-break-upon-gating" model of Mg^2+^ transport, where conformational flexibility and symmetry-broken fluctuating pentamer structures are required for the conduction. The work represents a tour-de-force, both in terms of the mechanistic insights and from the technical standpoints; the quality of the data is superb, and I anticipate that this paper will be highly impactful" and (2) "overall, the research is well-executed, and the spectroscopic data are in agreement with the MD simulations. The latter explains how the unligated state of CorA is challenging to crystallize. The paper is of broad interest, and the combination of these biophysical techniques is compelling" point to the interest in the paper. Both reviewers recommend a number of improvements to your manuscript prior to publication that do not involve additional experimentation.

Major Points

A) The paper is centered on the hypothesis that ion transport is dynamically driven. The authors have carried out extensive MD simulations to sample the different conformations of CorA. However, there are no indications in this paper of how the actual transport mechanism can occur. What happens to the ion-binding site during the interconversion between different states?

B) Most of the discussion is quite speculative (and long). The paper would benefit from shortening the discussion and may focus more on the findings of the current manuscript. For instance, it is not clear to this reviewer whether asymmetric conformations are partially competent for ion transport. Also, what is the evidence for the existence of a deep minimum for a well-defined open state (Figure 6)? Is it possible that these partially open states are sufficient to justify transport? Again, a figure, or better, the analysis of what happens to the binding site would probably explain the basis for ion transport.

C) The fitting with a mono-exponential function for the T1rho data points (F306) seems not to accurately report the decay of the signal (see panel D in Figure 5). Is it due to the presence of multiple states? Or is it due to the poor S/N in the spectra? Indeed, this reviewer understands the challenges for these kinds of systems, but the errors in the measurements at short delay times are quite large.

---

## [Author Response]

Major PointsA) The paper is centered on the hypothesis that ion transport is dynamically driven. The authors have carried out extensive MD simulations to sample the different conformations of CorA. However, there are no indications in this paper of how the actual transport mechanism can occur. What happens to the ion-binding site during the interconversion between different states?

We thank the reviewers for their comments and suggestions, and hope that this aspect is addressed more clearly in our revised version.

First of all, we would like to stress that our experiments do not provide the mechanistic details of Mg^2+^ transport. The SANS data does not have the required resolution and in NMR, well-resolved peaks could not be obtained for residues in the M1 binding site or in most of the stalk helix, preventing a direct measure of coupling between Mg^2+^ binding and increased dynamics in the TMD.

The primary goal of the MD simulations was to refine weighted populations of CorA pentamers that could explain the SANS data. To achieve this, we ran a biased (and coarse grained) simulation towards highly asymmetric states. The bias is enforced on residues in the intracellular domains (residues on the stalk helix, α7), which naturally produces structures where the residues of the M1 binding site (D89 on α3 and D253 on the neighbouring α7) come far apart. Whereas the fully Mg^2+^-bound crystal structure has M1 distances (Cγ to Cγ) ranging from 5.7 to 6.4 Å, the MD-generated clusters have distances of approximately 6 Å between only two of the protomers. For the remaining pairs, distances vary between 10 and 56 Å. Coordination of hydrated Mg^2+^, which has a radius of 4.8 Å (Maguire and Cowan, BioMetals 15: 203–210, 2002), is unlikely when distances between the two coordinating carboxyl groups of D89/D253 are much longer than 6 Å. As such, the MD-generated structures can potentially only bind one single hydrated Mg^2+^.

In the structure(s) generated by normal-mode analysis, which fit(s) the SANS data even better than the reweighted ensemble from MD, distances are retained around 6 Å in four M1 sites, whereas a single site has a distance of 64 Å. This overly large symmetry break in the intracellular region appeared unrealistic in the light of both our NMR data and previous CryoEM and crystallographic work as we discuss in the Results section of the article.

However, we have also now estimated the pore radii of these structures. In all cases, the pore is too narrow for hydrated Mg^2+^ to readily pass through. While this means that we cannot directly propose a structural basis for the transport mechanism, it highlights that asymmetric structures are not necessarily conducting. This is in line with our direct observation from SANS that the average solution structure of CorA is the same with and without Mg^2+^ present. As such, our work contributes to the understanding of CorA by the observation of asymmetric fluctuating states, even in the presence of Mg^2+^.

We now mention all these aspects in the Results section on page 11.

To obtain additional insight into the direct transport mechanism, more detailed MD simulations would be required. First, MD simulations should be conducted both with and without Mg^2+^ present. Second, in order to observe movement and detailed interactions of the Mg^2+^ ions, one would need to use all-atom MD simulations rather than the coarse-grained simulations that we use to observe the large structural changes. (We note that deriving accurate force fields for Mg^2+^-ions that allow for exchanging interaction partners is not easy, see e.g.: https://doi.org/10.1021/acs.jpcb.6b09262 and https://doi.org/10.1021/acs.jctc.0c01281). Fourth, the collective variable that we used to enhance the sampling was chosen with the goal of explaining the SANS data and overall conformation of CorA. Other collective variables, or combinations of collective variables, are likely required in order to optimize the simulation towards open states. Thus, substantially more simulation work would be required to actually see ion transport, and we think that this is out of the scope of our work here.

B) Most of the discussion is quite speculative (and long). The paper would benefit from shortening the discussion and may focus more on the findings of the current manuscript. For instance, it is not clear to this reviewer whether asymmetric conformations are partially competent for ion transport. Also, what is the evidence for the existence of a deep minimum for a well-defined open state (Figure 6)? Is it possible that these partially open states are sufficient to justify transport? Again, a figure, or better, the analysis of what happens to the binding site would probably explain the basis for ion transport.

Overall, we have reshaped the discussion and centred it on the actual results, so to put our novel findings in a better focus.

As for Figure 6, it provides exclusively a qualitative summary of the transitions which accompany variations in [Mg^2+^]. Neither the depth of the minimum nor the fine details of the conformational energy were experimentally determined in this work, but we realised that the previous graphic representation was misleading to this respect. We have re-sketched the figure, by taking multiple open conformations into account and highlighting that it is not the absolute value but the change in the energy profile that plays a role in the conformational transition.

As for whether the asymmetric conformations are competent for ion transport, or whether partially open states are sufficient to justify transport: as explained above, we have now analysed in deeper detail the computationally-obtained structures, which systematically feature pores too narrow to accommodate hydrated Mg^2+^. We believe that this is an indirect evidence that asymmetric structures are not necessarily conducting. Our conclusions are that such asymmetric structures are not partially open, but rather closed. This aspect should now emerge more visibly from the revised discussion.

Clearly, it would be interesting to extend our work with more sophisticated MD combined with experimental data as also discussed under point A. This however requires several new developments on both the NMR experimental and MD side.

C) The fitting with a mono-exponential function for the T1rho data points (F306) seems not to accurately report the decay of the signal (see panel D in Figure 5). Is it due to the presence of multiple states? Or is it due to the poor S/N in the spectra? Indeed, this reviewer understands the challenges for these kinds of systems, but the errors in the measurements at short delay times are quite large.

The reviewer is right, the exact quantification of site-specific relaxation rates from the experimental relaxation decays is undermined by the poor S/N in the spectra. We did try alternative functions such as biexponential curves, which fit the data with comparably poor agreements. Given this sensitivity limitation, however, our main point is that the remarkable variation in the relaxation decay profiles upon removal of Mg^2+^ can be appreciated even despite the intrinsically low signal-to-noise ratio of the experiments. We have now made this point explicit on page 13.